



# Causal Analysis of Aerosol Impacts on Isolated Deep Convection: Findings from TRACER

Dié Wang[1], Roni Kobrosly[2], Tao Zhang[1], Tamanna Subba[1], Susan van den Heever[3], Siddhant Gupta[4], and Michael Jensen[1]

[1]Brookhaven National Laboratory, Upton, NY 11937
[2]Icahn School of Medicine at Mount Sinai, New York, NY 10029
[3]Colorado State University, Fort Collins, CO 80523
[4]Argonne National Laboratory, Lemont, IL 60439

**Correspondence:** Dié Wang (diewang@bnl.gov)

**Abstract.** This study employs a novel application of causal machine learning, specifically g-computation, to quantify aerosol effects on deep convective clouds (DCCs). Focusing on isolated DCCs in the Houston-Galveston region, we leverage comprehensive ground-based observations from the TRacking Aerosol Convection interactions ExpeRiment (TRACER) to estimate aerosol influences on convective core depth, intensity, and area. Our results reveal that greater aerosol number concentrations

generally have a limited impact on convective core echo top height (ETH), with an increase of about 1 km (13% of average ETH). This effect is observed under specific conditions, particularly when ultrafine particles are activated in updraft regions. Additionally, greater aerosol levels correspond to increased convective core intensity and area, though these changes remain within radar measurement uncertainties. In DCCs associated with sea breezes, aerosol effects are more pronounced, resulting in a 1.4 km deepening of ETH. However, this heightened effect could be attributed to the exclusion of key confounders such as

boundary layer updrafts in the causal model. This study pioneers the application of causal machine learning to explore aerosol-convection interactions, shedding light on unraveling complex interplay between aerosols and meteorological variables.

## 1 Introduction

Deep convective clouds (DCCs) play a crucial role in the Earth's water cycle, as they generate a significant amount of global precipitation (e.g., Tan et al., 2015; Feng et al., 2016), regulate the global energy cycle through latent heat release (e.g., Tao et al., 2010), and vertical mass transport (e.g., Wang et al., 2019; Gupta et al., 2024), thereby driving large-scale atmospheric

circulations that impact climate sensitivity (e.g., Sanderson et al., 2008; Del Genio, 2012). Despite their significance for weather





and climate, accurately simulating DCCs in state-of-the-art numerical models remains challenging (e.g., Wang et al., 2020a; Prein et al., 2021; Wang et al., 2022b). Even fundamental convective characteristics such as updraft strength, cloud top height, and anvil mass detrainment, and the variations of these attributes over the diurnal cycle are difficult to simulate (e.g., Moncrieff, 2010; Bony et al., 2016). While field campaign data analyses have provided valuable insights into DCC processes (e.g.,

Polavarapu and Austin, 1979; Dye et al., 2000; Long et al., 2011; Chi et al., 2014; Barth et al., 2015; Jensen et al., 2016; Martin et al., 2017; Geerts et al., 2017; Varble et al., 2021; van den Heever et al., 2021; Jensen et al., 2022; Reid et al., 2023; Kollias et al., 2024), conventional model-observational validations mostly rely on bulk precipitation characteristics and/or sparse cloud dynamics observations from a small set of cases, thus offering only a limited understanding of the processes involved. Furthermore, case studies, by their nature, are confined to specific geographical regions, restricting model assessments to specific

environmental forcing conditions (e.g., Prein et al., 2022; Ramos-Valle et al., 2023).

Effective radiative forcing due to aerosol-cloud interactions stands out as the primary source of uncertainty associated with anthropogenic impacts on the climate system, according to the Intergovernmental Panel on Climate Change Sixth Assessment Report (IPCC, 2023). The challenging nature of accurately simulating aerosol-cloud interactions for DCCs was evidenced in a recent model intercomparison project (MIP) conducted by the Deep Convective Working Groups of the Aerosols, Cloud,

Precipitation and Convection (ACPC) initiative. This MIP was the first of its kind to assess the range of DCC sensitivity to aerosol loading across a suite of state-of-the-art convective system resolving models (van den Heever et al., 2018). Analysis of this suite of simulations conducted by Marinescu et al. (2021) focused on aerosol-induced changes to the terms in the vertical velocity momentum equation under prescribed low and high number concentrations of cloud condensation nuclei (CCN) conditions for a DCC case. This study showed substantial variability among the models in terms of the sensitivity of

precipitation amount and updraft velocity to aerosol loading. The significant differences among the various models highlight an urgent need to resolve the lack of convergence in aerosol-DCC interaction process representations within such high-resolution modeling frameworks.

Numerous studies have aimed to shed light on the complex nature of aerosol-DCC interactions, towards improving their representations in the models, sparking the description of a number of different underlying physical mechanisms (e.g., Andreae

et al., 2004; Khain et al., 2005; van den Heever et al., 2006; Rosenfeld et al., 2008; Lebo and Seinfeld, 2011; Li et al., 2011; Fan et al., 2018; Nishant et al., 2019; Grabowski and Morrison, 2020; Abbott and Cronin, 2021). The leading mechanisms include: (1) "cold-phase" invigoration, where high aerosol number concentrations, acting as CCN, nucleate more cloud droplets delaying hydrometeor growth via reduced collision-coalescence, lofting more liquid water above the freezing level, enhancing the total latent heating associated with freezing, increasing the buoyancy of rising convective parcels and ultimately invigorating

convective updrafts (e.g., Khain et al., 2005; van den Heever et al., 2006; Rosenfeld et al., 2008); (2) "warm-phase" invigoration, where high aerosol number concentrations nucleate more cloud droplets and reduce supersaturation with respect to liquid water, increasing latent heat release through additional condensation of water vapor, and invigorating convective updrafts (e.g., Lebo, 2018; Fan et al., 2018, 2020); and (3) "humidity-entrainment" invigoration, where high aerosol number concentrations increase the environmental humidity by producing clouds that detrain more condensed water into the surrounding air, leading

to higher humidity that favors large-scale ascent and stronger convective updrafts (Abbott and Cronin, 2021). This wide range



of plausible mechanisms highlights the challenge of constraining this important problem with current observations. The lack of clear understanding further underscores the need for more robust and high-resolution observational data along with the development of advanced statistical methods and modeling frameworks that can better elucidate the complexity of aerosol-DCC interactions.

Despite a range of hypothetical mechanisms for aerosol-DCC invigoration, recent studies continue to challenge the importance of these mechanisms relative to other DCC forcing (e.g., Grabowski and Morrison, 2020; Igel and van den Heever, 2021; Dagan, 2022; Romps et al., 2023; Peters et al., 2023). From an observational perspective, this challenge arises, in part, from a lack of key supporting observations of vertical velocity, hydrometeor microphysical properties, and water vapor supersaturation within the convective core regions of DCCs, all of which would assist to provide further clarity on the invigoration processes.

Moreover, the thermodynamic and kinematic regimes under which aerosol-DCC interactions may be significant remain unresolved. Quantifying aerosol impacts on DCCs is further complicated because small-scale perturbations in large-scale vertical velocity, relative humidity, and other meteorological factors, such as wind shear and atmospheric instability, can potentially affect DCC intensity in a manner comparable to aerosol-induced changes (e.g., Fan et al., 2009; Storer et al., 2010; Grant and van den Heever, 2015; Lebo, 2018; Dagan et al., 2020; Park and van den Heever, 2022). Disentangling aerosol impacts on

DCCs from those driven by meteorological factors is therefore difficult (e.g., Varble et al., 2023).

To accurately assess the contribution of aerosols to DCC properties, a variety of techniques and methods have been developed from both modeling and observational perspectives. On the modeling side, a range of statistical methods and modeling frameworks have been established, including the simple factor separation approach (van den Heever and Cotton, 2007; Grant and van den Heever, 2014), more sophisticated statistical emulators (Lee et al., 2011; Johnson et al., 2015; Wellmann et al.,

2018; Park and van den Heever, 2022), and the piggybacking approach (Grabowski, 2015). These techniques have achieved some success in separating aerosol effects from the impacts of other forcing factors on DCC development. However, from the observational side, achieving this separation remains a longstanding challenge. The majority of observational studies have used multivariable models or basic machine learning approaches to mitigate potential confounding bias arising from meteorological covariates (e.g., Li et al., 2011; Storer et al., 2014; Veals et al., 2022; Zang et al., 2023). Nevertheless, it is important

to emphasize that these methods can merely estimate the association or correlation between aerosols and DCCs, and proving correlation does not imply causation. Therefore, to gain a more thorough understanding of the underlying causal relationships and effects of aerosols on DCCs - or the absence of such effects - advanced statistical techniques are essential. Furthermore, it is vital to employ comprehensive, high-resolution observations of DCCs and aerosols to capture these intertwined physical processes and identify potential "fingerprints" of aerosol-DCC invigoration.

This study presents a novel statistical investigation into the aerosol-DCC interactions by applying causal machine learning methods to comprehensive observational datasets. The goal is to provide observational evidence of the aerosol casual effects on DCC intensity (either invigoration or enervation). The datasets we used were collected during the TRacking Aerosol Convection interactions ExpeRiment (TRACER; Jensen et al., 2023) in Houston-Galveston, operated by the Department of Energy (DOE) Atmospheric Radiation Measurement (ARM) Climate Research Facility (Mather and Voyles, 2013). We focus

on DCCs occurring during the summer months in 2022 from June to September, the TRACER Intensive Operational Period



(IOP), as the synoptic conditions show less variation during this time interval and additional measurements of cloud, aerosol, and atmospheric profiling are available. To address the challenge of untangling the effects of aerosols from meteorological variables and estimating the aerosol causal effects, we employ a well-established causal model, g-computation (Robins, 1986), in combination with a Self-Organizing Map (SOM) approach (Kohonen, 1990). The SOM is used to identify synoptic regimes

conducive to isolated DCCs, thereby minimizing the impact of large-scale ascents on their interactions with aerosols (Wang et al., 2022a). G-computation is chosen since it stands as a powerful model that facilitates the estimation of causal effects and exhibits versatility in handling a broad spectrum of sample sizes, making it particularly well-suited for studies with a limited sample size (e.g., Le Borgne et al., 2021). Furthermore, its flexibility allows us to model the relationship of interest using the statistical models of our choice (e.g., Chatton et al., 2020). This study marks the first application of the g-computation model

to investigate aerosol-cloud interactions.

## 2 Instrumentation and Datasets

### 2.1 DCC properties

As the first step in the investigation, we employ a Lagrangian framework to detect the formation and propagation of DCC rainfall cores and quantify their convective characteristics throughout their lifecycle. The term "DCC rainfall cores" typically

refers to the convective regions in DCCs with heavy rainfall rates at the surface with a maximum value exceeding 10 mm/hr (e.g., Wang et al., 2018; Zhang et al., 2021). The maximum height of these cores can serve as a proxy for updraft strength, as it correlates closely with the ability of updrafts in convective regions to lift large hydrometeors to higher altitudes, resulting in deeper convective systems (e.g., Heymsfield et al., 2010; Liu and Zipser, 2013; Guo et al., 2018).

More specifically, we tracked the trajectory of DCC rainfall cores using TINT (TINT Is Not TITAN [Thunderstorm Iden-

tification, Tracking Analysis and Nowcasting; Dixon and Wiener, 1993), a convective cell tracking algorithm developed by Raut et al. (2021). Building upon our prior research (Wang et al., 2024), we have effectively used this algorithm to analyze the level-II data (NOAA, 1991) from the S-band Doppler weather radar KHGX-Houston at 1-km horizontal resolution within a domain of 400 km × 400 km centered around the radar location (Figure 1). As a result, we have generated a comprehensive tracked DCC rainfall core dataset for the TRACER IOP during the summer of 2022 (Wang et al., 2024).

In that study and the current one, DCC rainfall cores are defined using radar observations as contiguous areas where the 2-km radar reflectivity (Z) is greater than 10 dBZ, the lower limit for rain echo detection by NEXRAD radar systems, and the maximum 2-km Z value exceeds 40 dBZ (Anagnostou, 2004; Moroda et al., 2021). Note that different reflectivity thresholds varying from 30 to 40 dBZ have been selected for studying DCC convective cores in various climate conditions, depending on the objectives of the studies (e.g., Giangrande et al., 2023; Gupta et al., 2024). Additionally, these cores must exhibit a

30-dBZ echo top height (ETH) exceeding 5 km above ground level at some point during their lifetime to exclude the presence of shallow convection, aligning with a similar definition proposed by Dixon and Wiener (1993). Further details regarding additional criteria for identifying and tracking these rainfall cores can be found in Texts S1 and S3 in the supporting information and in Wang et al. (2024).





Note that the first identification of the DCC rainfall core using the tracking algorithm signifies the initiation of surface rain-
fall associated with the DCC core. The tracking algorithm can no longer identify the core once the DCC ceases to produce
moderate precipitation (maximum 2-km $Z < 40$ dBZ), marking the termination of the convective stage. In other words, the
tracked lifetime of the cores excludes the initiation stage of non-precipitating cumulus clouds and the dissipation stage of
non-precipitating anvil clouds. Given the defined criteria and these important caveats, Figure 1 shows the locations of tracks
corresponding to DCC rainfall cores formed within a 20 km radius of the TRACER main site (M1) at La Porte, TX, where
comprehensive aerosol and meteorological measurements were collected. Additionally, Figure S1 (in the supporting informa-
tion) shows similar plots for DCCs identified within 30 km, 40 km, and 50 km radii of the M1 site. Table 1 details the number
of DCC rainfall cores tracked and considered in each scenario.

The DCC intensity is quantified using the maximum 30-dBZ ETH of the tracked core as the primary indicator (e.g., Liu and
Zipser, 2013; Guo et al., 2018). Additionally, some studies have used the maximum height of the 10-dBZ or 15-dBZ echo as
proxies for cloud depth and convective updraft strength (e.g., Hu et al., 2019; Veals et al., 2022). Therefore, to test the sensitivity
of the results to our assumed proxy, we also consider the maximum 15-dBZ ETH, calculated using the KHGX-Houston radar
data, as a secondary indicator of convective intensity.

## 2.2   Meteorological variables

Meteorological conditions are crucial in shaping the formation and evolution of DCCs and may co-vary with aerosol properties,
complicating the accurate quantification of aerosol-DCC interactions (e.g., Lee et al., 2008; Storer et al., 2010; Grant and
van den Heever, 2015; Lebo, 2018; Dagan et al., 2020; Park and van den Heever, 2022; Zang et al., 2023; Varble et al., 2023).
These meteorological variables or convective indices, influencing both aerosol activation and convective updraft strength, are
termed "confounders" or "confounding variables" in causal inference (Jesson et al., 2021), and introduce the potential for
spurious associations.

The convective indices analyzed in this study include convective available potential energy (CAPE), lifting condensation
level (LCL), level of neutral buoyancy (LNB), environmental lapse rate (ELR) between 3 km and the surface ($ELR_3$), ELR
between 6 km and 3 km ($ELR_6$), low-level vertical wind shear from the surface to 5 km (LWS), and low-level mean relative
humidity below 5 km (RH). These variables have been identified in previous studies as the most influential meteorological
factors altering the impacts of aerosols on convective updrafts and precipitation because these factors regulate the kinematic
and microphysical processes in DCCs and the kinematic-microphysical feedback (e.g., Khain et al., 2008; Khain, 2009; Nishant
et al., 2019; Fan et al., 2009; Tao et al., 2012; Storer et al., 2010, 2014; Varble, 2018; Wang et al., 2020a; Veals et al., 2022;
Sun et al., 2023; Masrour and Rezazadeh, 2023).

To quantify these convective indices, measurements from the ARM balloon-borne sounding system (SONDE) launched at
the M1 site are used. Radiosondes were typically launched four times a day at approximately 0530, 1130, 1730, and 2330 UTC
during the TRACER campaign, with additional launches at 1900, 2030 and 2200 UTC on enhanced operational days (as listed
in Table S1, in the supporting information). These radiosondes provide in situ measurements of atmospheric thermodynamic
state profiles, wind speed, and wind direction. To address the sensitivity of these variable calculations to the choice of initial





parcel conditions, three scenarios are considered. These scenarios involve lifting different air parcels to initiate a convective cloud: the surface-based parcel (*sfc*), the most unstable parcel (*mu*), and the mixed-layer parcel (*mix*). Detailed information on these calculations can be found in Wang et al. (2020b).

Note that, in addition to the convective indices mentioned above, other factors such as entrainment rate (Abbott and Cronin, 2021; Peters et al., 2023) may also be important in regulating the aerosol-DCC interactions; however, no direct measurements of these quantities are available from TRACER. Therefore, these factors are not included in the analysis. The potential biases in the quantification of the aerosol causal effects due to these exclusions will be discussed in Section 4.6.

## 2.3 Surface aerosol measurements

The Aerosol Observing System (AOS; Uin et al., 2019) within the ARM mobile facility (AMF; Miller et al., 2016) was used for *in situ* aerosol measurements at the surface.

The dual-column CCN counter (Column A and Column B) was used to determine CCN number concentrations ($N_{ccn}$). This instrument measures the number and size of activated aerosol particles for each column at a specific supersaturation (SS) level. Particle size, after humidification, can be measured between 0.75-10 $\mu$m, and the range of particle number concentration measurement depends on the SS caused by the growth kinetics of activated particles. Column A has varying SS setpoints between 0% and 1% at a frequency of 1.5 hours, while Column B has a fixed SS setpoint of 0.35%. Due to the unavailability of Column B data at the time of the study, only Column A data were considered. The dataset used includes the number concentration of CCN at SS setpoints of 0.1%, 0.2%, 0.4%, 0.6%, 0.8%, and 1%, which are referred to as $N_{ccn01}$, $N_{ccn02}$, $N_{ccn04}$, $N_{ccn06}$, $N_{ccn08}$, and $N_{ccn1}$, respectively. Note that these measurements were bias-corrected based on a CCN closure study using methods developed by Petters and Kreidenweis (2007). As direct measurements of SS in convective cloud updrafts are not available (i.e., updraft SS is unknown), we consider all six parameters as potential predictors (individually) in the causal model.

Moreover, the total aerosol number concentrations including ultrafine particles in the nucleation and Aitken mode along with larger, accumulation mode aerosols are considered. The total aerosol number concentrations have the potential to influence DCC evolution, assuming that these particles may be activated as CCN in DCC updrafts in which a range of SS values may be present (e.g., Politovich and Cooper, 1988; Benmoshe, 2010). These quantities were measured by the condensation particle counter (CPC) installed as part of the ARM AOS (Singh and Kuang, 2024). Two types of CPC instruments were used in the AOS: ultrafine CPC instruments (CPCU) and fine mode CPC instruments (CPCF). The CPCU counts aerosol particles with diameters ranging from 3 to 3,000 nm ($N_{ufp}$), while CPCF counts aerosol particles with diameters ranging from 10 to 3,000 nm ($N_{cn}$).

These aerosol properties ($N_{ccn}$ at various SSs, $N_{cn}$, and $N_{ufp}$) were measured at a temporal resolution of 1 minute or less, while the radiosondes, used to derive the meteorological parameters, were launched four to seven times per day. To synchronize the two datasets, we employ two commonly used methods from previous studies to explore the sensitivity of results to the averaging process. One approach entails averaging the aerosol properties over a 1-hour period following the launch of a radiosonde (post-sounding averaging; e.g., Veals et al., 2022). The second method involves utilizing a 1-hour period preceding





the initial identification of the rainfall cores, representing the aerosol conditions before the detection of precipitation at the surface (prior-rain averaging).

Based on a two-sample t-test (Welch, 2005), the differences between the distributions of the aerosol properties derived using the post-sounding averaging and the prior-rain averaging are statistically insignificant. This is true for all aerosol properties considered in this study. In addition, the median values of the aerosol parameters from these two averaging methods are also comparable, with relative differences ranging from 2% ($N_{ccn02}$) to 23% ($N_{ccn01}$). Similar results are found when comparing the variability of aerosol properties within the 1-hour averaging period, showing a consistent median value of the standard deviation for these parameters across all the DCC samples.

## 2.4 Pairing environmental variables with tracked DCCs

In order to establish causal relationships between aerosol and DCCs and facilitate calculations using g-computation, we align environmental variables (aerosol and meteorology) with tracked DCC properties. This is achieved by identifying DCC rainfall cores that form within 6 hours after launching each sounding, within a maximum distance of 50 km from the M1 site. The DCC tracking results are then averaged to represent the mean DCC properties for each corresponding sounding. The specifics of the number of samples are detailed in Table 1.

The choice of a 6-hour time gap and a 50 km distance threshold as the upper limit ensures the description of environmental conditions in close proximity in space and time to the early stages of DCC rainfall cores. This decision is informed by the relative stability observed in convective parameters, such as CAPE, at a specific location within six hours before the formation of DCC precipitation (e.g., Prein et al., 2022). Additionally, given that the M1 site is located approximately 51 km away from the Gulf of Mexico coastline, the 50 km threshold is intended to exclude DCCs initiated over the Gulf of Mexico, thereby limiting, to a large extent, the confounding influence of maritime conditions on DCC initiation.

However, to account for the heterogeneous and evolving nature of meteorological conditions that may impact DCC development, we evaluate the spatial and temporal scales of meteorological influences on DCC rainfall core characteristics through sensitivity tests. These tests involve examining DCC rainfall cores initially identified within a radius of 20 km, 30 km, 40 km, and 50 km from the M1 site, considering two different groups of soundings: those launched within 4 hours and those within 6 hours before the initial identification of the DCC rainfall cores.

Given the temporal and spatial constraints of the current observations, the purpose of these tests is to strike the best possible balance between accurately characterizing the initial conditions where DCCs are embedded with the available observations and maintaining a sample size that optimizes the performance of the causal model.

Note that various pairing methods have been used in prior observational studies on aerosol-DCC interaction with the goal of expanding the sample size. One approach involves searching for a sounding launch within a specific time period preceding the identification of each tracked DCC within a defined domain (Veals et al., 2022). This increases the number of samples to be equal to the number of tracked DCCs, in contrast to our original method where the number of samples is equivalent to the number of sounding launches. It is crucial to acknowledge that different DCCs may correspond to the same sounding profiles in the Veals et al. (2022) method, limiting the natural variability of the pre-convection environment across different cases.





Additionally, this lack of variability violates one of the assumptions of the g-computation model (i.e., stable unit treatment value assumption), which will be detailed in Section 3.4. Therefore, the subsequent analysis is exclusively conducted on datasets generated from the original method of using the mean properties of DCC rainfall cores tracked within 4 or 6 hours after the launch of each sounding.

## 3  Aerosol-DCC Causal Framework

To assess the potential impacts of aerosols on DCC ETH, a framework consisting of a three-step methodology is developed, incorporating the use of SOM and the g-computation model, as depicted in Figure 2.

In the first step (Figure 2a), we use the SOM method (Section 3.1) to classify the synoptic weather regimes with the aim of singling out DCC cases occurring within the context of weak synoptic-scale forcing. This choice serves to mitigate the

235 potential influence of large-scale ascent on the evolution of DCCs. This is important given our focus on assessing aerosol effects on convective and buoyancy-driven updrafts, rather than on synoptic-scale, meteorological contributions to the convective updrafts. The characteristics of the synoptic regimes over the Houston-Galveston region, details of the SOM setup, training process, and further information on the SOM method can be found in Wang et al. (2022a).

The second step (Figure 2b) involves the preparation of data for the g-computation model, with a focus on determining

the exposure, confounding, and outcome variables (Sections 3.2 and 3.3). The terms "exposure" and "outcome" refer to the central variables of interest, where the exposure is believed to causally influence the outcome, and we aim to estimate this effect. If a variable affects both exposure and outcome variables, it is called a "confounder" or "confounding variable". In the context of this study, the potential exposure variables refer to aerosol properties such as $N_{ccn}$ at various SSs, $N_{cn}$, or $N_{ufp}$, while the outcome variables refer to DCC properties such as 30-dBZ ETH or 15-dBZ ETH. The confounding variables refer to

convective indices that may influence both DCC and aerosol properties such as CAPE.

Finally, in the last step (Figure 2c), the g-computation model is used to estimate the aerosol causal effects on DCC ETH (Section 3.4). This analysis involves predicting the outcome variables under different hypothetical scenarios wherein the relevant confounding variables are held constant. Following the established terminology, these scenarios are hereafter referred to as counterfactual states (e.g., Hernan, 2004; Naimi et al., 2017). Specifically, we consider two counterfactual states: the polluted

state, assuming that all DCCs are exposed to aerosols, and the clean state, assuming that none of the DCCs are exposed to aerosols. By comparing the predicted DCC ETH between these two counterfactual states, we can provide unbiased estimates of the causal effects of aerosols on DCC ETH.

### 3.1  Minimizing the influence of variability in synoptic-scale forcing

In this subsection, we describe the first step in which we use an unsupervised machine learning technique, SOM, to categorize

synoptic weather patterns in the Houston-Galveston area (Figure 2a). The purpose of this step is to focus on DCC-aerosol relationships while minimizing the influence of synoptic-scale ascents such as those that are associated with strong synoptic-scale troughs and onshore winds.





In our prior studies (Wang et al., 2022a), we identified three main synoptic patterns in the Houston-Galveston region using the SOM approach, including a pre-trough, a post-trough, and an anticyclonic regime. The input data for SOM were 700-hPa geopotential height anomalies (recorded at 0000 UTC) from the European Centre for Medium-Range Weather Forecasts (ECMWF) Reanalysis version 5 (ERA5; Hersbach et al., 2020) during the summer months (June to September) of 2010 to 2022. Among these regimes, the anticyclonic regime is the most frequent, representing 49% of all days across the 13-year dataset, occurring predominantly in July and August. The corresponding regime for each day during the TRACER IOP can be found in Figure S2 (in the supporting information).

During the anticyclonic regime, a high-pressure system typically resides over the Houston-Galveston area as the Bermuda High has shifted toward the west. The region is positioned on the inner eastern edge of a ridge at 500 hPa, and on the inner western edge of an anticyclonic system at 850 hPa. This configuration creates a stable synoptic background characterized by large-scale subsidence over the study area, weak horizontal winds throughout the troposphere, and moderate column water vapor content (Wang et al., 2022a). These conditions are favorable for the formation of locally-forced, isolated DCCs with minimal wind shear and moderate low-level moistening (Wang et al., 2024). As such, this environment is conducive to studying the interactions between DCCs and aerosols. Conversely, the pre-trough and post-trough regimes are associated with large-scale trough intrusions and moisture transport from the Gulf of Mexico, which are more likely to promote organized convective clouds over the region that are primarily driven by large-scale dynamics (Wang et al., 2022a). Therefore, these specific cases are excluded from our study, aligning with our emphasis on evaluating aerosol impacts on buoyancy-driven DCCs, which are comparatively less influenced by the large-scale ascent.

## 3.2 Determination of exposure variables

This subsection outlines the determination of exposure variables for the g-computation model. Given our potential exposure variables or predictors, namely aerosol or CCN number concentrations at various SS levels, we aim to identify the most relevant aerosol parameters, impacting the DCC ETH. In other words, the selected exposure variables are those that demonstrate a significant association with DCC ETH.

To achieve that, we evaluate the performance of a simple linear regression (SLR) model when attempting to predict DCC ETH using each aerosol parameter individually. The *P*-value of each SLR model is assessed, indicating the statistical significance of the associations between ETH and aerosol parameters.

Note that since the g-computation calculation in the next step requires a binary exposure variable, the aerosol parameters are transformed into a binary distribution (0 or 1). Cases with $N_{ccn}$ at various SSs, $N_{cn}$, or $N_{ufp}$ above the median value are categorized as polluted cases with a scaled value of 1, while cases below the median value are classified as clean cases with a scaled value of 0. This transformation also helps address potential biases associated with the $N_{ccn}$ measurements/calculations during TRACER since the exposure state is defined relative to a bulk statistical parameter (median value) which minimizes the dependence on individual measurement uncertainty. Several tests have been conducted to assess the sensitivity of the results to the clean-polluted separation threshold by using aerosol number concentrations less than the $40^{th}$ percentile for clean and higher than the $60^{th}$ percentile for polluted. The valid models remain the same, and similar causal effects are shown compared





to the original setting. For simplicity, the following analysis uses the median value for the separation and the other thresholds are not shown.

Figure 3 illustrates the *P*-values resulting from each fitted model, with 30-dBZ ETH/15-dBZ ETH as the outcome variable
and each aerosol parameter ($N_{ccn}$ at various SSs, $N_{cn}$, or $N_{ufp}$) as the predictor (derived from the two averaging methods described above). A *P*-value below 0.05 signifies a statistically significant association between the predictor and outcome variables. Only predictors demonstrating a *P*-value below 0.05 are considered as exposure variables in the subsequent causal analysis.

The most notable feature from Figure 3 is that only a small fraction (20 out of 128, accounting for 16%) of the SLR models
are statistically significant. This result suggests that, in the majority of scenarios, the aerosol number concentration is not a reliable influencer of changes in DCC ETH, suggesting limited favorable conditions for aerosol loading to impact DCC updraft strength. Among the valid SLR models ($P < 0.05$), most of them use $N_{cn}$ or $N_{ufp}$ as the predictor variable. This implies that aerosol loading potentially influences convective updraft intensity if all particles are activated in those updrafts, including the ultrafine particles. Whether a certain level of SS or a range of SS values can be reached within those updrafts to activate all the
aerosol particles is not observed. The actual SS values may depend on meteorological conditions such as atmospheric instability and moisture availability, in addition to updraft strength. This underscores the importance of considering these meteorological conditions (like the convective indices defined above) as confounding variables in the causal model.

### 3.3 Identification of confounders

In this subsection, our primary objective is to identify a specific set of confounding variables from a range of convective indices
introduced in Section 2.2. To achieve this, we assess the Pearson correlation coefficients (*R*) between 30-dBZ ETH/15-dBZ ETH and selected convective indices, as delineated in Table 2 and shown in Figure 4.

Positive *R*-values between 0.2 and 0.4 are evident when examining the relationship between LNB, $CAPE_{mu}$, LCL, or $ELR_3$ and 30-dBZ ETH. The positive association between $CAPE_{mu}$ and 30-dBZ ETH can primarily be attributed to the direct impact of CAPE on the maximum potential velocity of updrafts, independent of entrainment and hydrometer loading effects
(Weisman and Klemp, 1984; Kirkpatrick et al., 2011). This relationship finds support in observations across diverse climate regions, including Darwin, Australia (Kumar et al., 2013), the Sierras de Córdoba mountain range (Veals et al., 2022), and the central Amazon (Wang et al., 2019). This robust association is also present when using surface parcels but diminishes with mixed-layer parcels (Table S2, in the supporting information). Additionally, LNB shows weak, positive correlation with both 30-dBZ ETH and 15-dBZ ETH, since it is highly correlated to CAPE (Figure 4).
Concerning LCL, its impact on 30-dBZ ETH can be explained by its previously demonstrated positive correlation with the width of updrafts at cloud base (McCaul and Cohen, 2002; Mulholland et al., 2021). In other words, a higher LCL tends to promote wider boundary-layer updrafts. These wider updrafts are more likely to evolve into expansive and deeper updraft cores within DCCs since they experience less dilution of buoyancy due to entrainment compared to narrower updraft cores. Consequently, this leads to a higher 30-dBZ ETH. Similarly, a steeper $ELR_3$ is closely linked to a higher LCL ($R = 0.9$, Figure
4), and subsequently, a higher 30-dBZ ETH. This steeper $ELR_3$ also corresponds to a "fatter" buoyancy profile (Zipser and





LeMone, 1980), where CAPE is concentrated at lower levels. An air parcel accelerates more rapidly through these levels, reducing the exposure time for entrainment and other processes (Wang et al., 2020b). Therefore, a higher 30-dBZ ETH may be expected.

LWS is another essential factor governing DCC updraft intensity and regulating aerosol-DCC interactions, particularly in 330 organized DCCs (e.g., Fan et al., 2009; Baidu et al., 2022). However, in the specific isolated DCC environment studied here, it has no association with 30-dBZ ETH, but does have a weak, negative correlation with 15-dBZ ETH. Therefore, LWS is excluded (included) as a confounding variable when the 30-dBZ ETH (15-dBZ ETH) is considered as the outcome variable in the causal model.

Overall, LNB, CAPE, LCL, and $ELR_3$ exhibit weak to moderate $R$-values across various scenarios, making them suitable 335 potential covariates for predicting 30-dBZ ETH alongside aerosol properties. However, high correlation is found between LNB and CAPE ($R = 0.9$, Figure 4) as higher values of CAPE indicate greater atmospheric instability, allowing air parcels to rise to higher altitudes, thus potentially higher LNB. Similarly, strong correlation is also exhibited between $ELR_3$ and LCL ($R = 0.9$, Figure 4), which can be attributed to their shared relationship with temperature variations in the lower atmosphere. To address multicollinearity concerns, only one variable from each pair is selected as a confounding variable, which can otherwise 340 lead to increased variance in estimated coefficients within the g-computation model. Further discussion on multicollinearity is presented in Text S4 of the supporting information. Note that opting for fewer confounders also has the advantage of partially addressing challenges related to the accuracy of the causal model given the relatively limited sample size in this study.

Finally, CAPE and $ELR_3$ are chosen due to their higher $R$-values with 30-dBZ ETH compared to their counterparts. Following a similar logic, CAPE and LWS are selected as confounders when the 15-dBZ ETH is used as the outcome variable in the 345 causal model. Moreover, these selected confounding variables exhibit a stronger association with aerosol parameters compared to other convective indices (Figure 4). Similar findings are reported in previous studies by Varble (2018). Using the surface and mixed-layer parcel, a consistent conclusion is drawn (Figures S3 and S4, in the supporting information).

## 3.4 G-computation causal model

G-computation, along with g-methods in general (Robins, 1986), is widely utilized across various fields and has garnered 350 significant attention in the scientific community for causal analysis, particularly in epidemiology (e.g., Mooney et al., 2021; Chatton et al., 2020). This model is a statistical technique utilized to estimate the causal effect of an exposure or condition in the presence of a set of confounders in observational studies.

The accuracy of the g-computation model relies on several key assumptions. These assumptions include:

1. Temporality: It assumes that the exposure occurs before the outcome. In our study, we use aerosol properties observed
prior to the detection of convective rainfall echos at the surface for all DCC cases, satisfying this requirement.

2. Stable unit treatment value: It assumes that the exposure of one observation to the exposure variable does not affect
    the potential outcomes of other observations. While the first initiated DCCs over a specific region may modify the
    environmental conditions for subsequent DCCs, our study primarily focuses on isolated DCC cases with short durations





and limited cloud to cloud interactions. The DCC cases also occurred on different days during the IOP, leading us to expect minimal impact of one DCC on another DCC.

3. Positivity: It assumes that there is sufficient variability in the exposure and outcome variables for each confounder in the data. Our dataset shows considerable variability in both aerosol and meteorological variables (Figures 5 and 7), which satisfies this assumption.

4. Ignorability: We assume that all major confounding variables are included in the data. Critical quantities known to influence ETH, such as CAPE and LWS, are explicitly included or discussed, sufficiently supporting this assumption. While variables like entrainment rate and vertical velocity also likely confound aerosol and DCC properties, their exclusion in this study is dictated by the absence of direct measurements during TRACER. To address potential biases arising from unobserved confounders, a comprehensive discussion is provided in Section 4.6.

The g-computation model consists of three steps used to estimate the causal effect of an exposure (Figure 2):

1. The outcome (Y, ETH in this case) is modeled as a function of the exposure (A, aerosol number concentration in this case) and relevant confounders (V; CAPE and $ELR_3$/LWS in this case) using a statistical model such as logistic, linear regression, or a predictive machine learning model ($E(Y|A, V)$), commonly known as the "Q-model".

2. The fitted Q-model is used to predict counterfactual outcomes for each observation under each exposure scenario (whether exposed to high concentration of aerosols or not). This is done by setting A = 1 (polluted) and subsequently A = 0 (clean) into the Q-model fit to obtain predicted outcomes for these two settings.

3. Finally, the average causal effect is calculated by taking the difference between the average counterfactual outcomes under the exposed and unexposed conditions.

We describe each step in detail in the following subsections.

### 3.4.1   Fit a Q-model

The first step in the g-computation process involves fitting a statistical Q-model to the dataset. Given the limited number of DCC cases in our study, we have chosen to use the multiple linear regression (MLR) model, which is suitable for analyzing relatively small datasets. In our case, the MLR model includes the outcome variable, Y, which represents the 30-dBZ (15-dBZ ETH), and the exposure variable, A, which represents $N_{cn}$ or $N_{ufp}$. We also include two confounding variables, V, which are V1 = CAPE, V2 = $ELR_3$ for Y = 30-dBZ ETH and V1 = CAPE, V2 = LWS for Y = 15-dBZ ETH.

The MLR model can be expressed as follows: Y = b0 + b1A + b2V1 + b3V2. Here, b0 represents the value of Y when all independent variables (exposure and confounding variables) are equal to zero, or it can be interpreted as the residual term. The coefficients b1, b2, and b3 are the estimated regression coefficients associated with the exposure and confounding variables.

To further address potential multicollinearity issues between the covariates, we perform standardization on all the confounding variables. This standardization process transforms the variables so that they have a mean of 0 and a standard deviation of 1.





It is achieved by subtracting the mean of each variable from each observation and then dividing by its standard deviation. This
procedure helps mitigate the impact of different units and ranges of the variables, allowing for meaningful comparisons of the
regression coefficients on the same scale.

One of the important steps is to evaluate the performance of the MLR model, which can lend confidence in the causal effect
estimated in the next step. This is achieved by examining the key assumptions (i.e., linearity, homoscedasticity, normality,

independence, and multicollinearity) of the MLR models as described in Text S4 in the supporting information. Overall, all
valid scenarios presented in Section 3.2 satisfy these assumptions.

### 3.4.2  Estimate counterfactual outcomes and average causal effects

The next step involves estimating the counterfactual outcomes under different conditions and calculating the average causal
effect of the aerosol number concentration on ETH.

First, we use the fitted MLR model to predict the ETH values under both clean and polluted conditions for each observation
in the dataset. This is done by setting the aerosol number concentration (exposure variable) to 0 (A=0, in the Q-model) for
clean conditions and 1 (A=1, in the Q-model) for polluted conditions for each observation in the data set, while keeping the
other confounders at their observed values.

Second, we calculate the causal effect by comparing the counterfactual predicted outcomes under the two aerosol conditions.

This involves taking the difference between the predicted ETH values under polluted conditions and clean conditions for
each observation. To estimate the average causal effect on ETH across the entire dataset, we weight these differences by the
proportions of observations in the polluted and clean groups.

## 4   Houston-Galveston environments and results from the causal model

In this section, we first provide an overview of the characteristics of the DCC properties and their associated aerosol and

410 meteorological conditions in the Houston-Galveston region. Then, we present results from the causal analyses and discuss
potential uncertainties of the results.

### 4.1   DCC properties and their associated environmental conditions

In Figure 5, we illustrate the distributions of selected convective indices representative of the pre-convective conditions. During
the selected DCC days, the influence of anticyclonic large-scale flow leads to moderate low-level moistening, resulting in

medium-to-high low-level RH (mean RH values below 5 km) of approximately 70% (Figure 5o). This moistening causes air
parcels to saturate quickly at lower levels when lifted, leading to a relatively low mean LCL of 1 km (Figures 5g-i), although
this value is higher compared to the LCL values in more humid conditions, such as an oceanic environment with a mean
low-level RH of 80% (Wang et al., 2020b). The LCL is in close proximity to the LFC, with a smaller median difference of
100 m when using the most-unstable parcel and a larger difference of 600 m when using a mixed-layer parcel (Figures 5j-l).

Consequently, the convective inhibition ($CIN_{mu}$) is relatively low, with a median value of -0.7 J/kg (not shown).





Under these conditions, (adiabatically) lifted parcels can ascend to significant heights, even reaching the tropopause, with a
mean $\mathrm{LNB}_{mu}$ of 14.6 km (Figure 5d). When considering mixed-layer parcels, the mean value of $\mathrm{LNB}_{mix}$ decreases to 13.9
km as expected (Figure 5f). This environment allows for the accumulation of significant $\mathrm{CAPE}_{mu}$ throughout the troposphere,
with a median value of approximately 3,407 J/kg (Figure 5a). There are limited changes in CAPE values when using the surface

parcel in the calculation compared to $\mathrm{CAPE}_{mu}$ (Figure 5b), which implies that most of the most-unstable parcels are from near
surface levels. Under such circumstances, using surface aerosol measurements to represent the in-cloud aerosol properties may
result in reduced uncertainty compared to applying the same assumption to study elevated DCCs. The LWS is relatively weak,
with a mean value of 5.7 $ms^{-1}$ (Figure 5n), compared to wind shear values that support the initiation of organized convective
systems (Baidu et al., 2022).

The distributions of convective properties associated with DCCs initiated under such meteorological conditions are illus-
trated in Figure 6. In this demonstration, the selected DCC cases are those identified within a 50 km radius from the ARM
M1 site. The definitions of these properties can be found in Text S2 in the supporting information. These tracked DCC rainfall
cores show intense rainfall rates, exhibiting a mean maximum 2-km $Z$ of 54 dBZ (Figure 6a). The maximum 30-dBZ ETH for
half of these cores extends above 7 km (Figure 6b). These cores are small in size, with their maximum area having a median

value of approximately 52 $km^2$ during their lifetime (Figure 6c), confirming their more isolated nature. Most of these rainfall
cores form in the afternoon hours with a peak in the number of cores initiating around 2000 UTC, corresponding to 1500 local
time (Figure 6d). This observation confirms that these cases are predominantly locally driven under weak synoptic-forcing and
influenced by surface heating and/or sea-breeze circulations (Wang et al., 2022a). It is therefore no surprise that these cores
propagate at a relatively slow speed of 5 $ms^{-1}$ (Figure 6e) and have a relatively short duration of less than an hour (51 min,

Figure 6f). The influence of aerosol number concentrations on these locally-driven DCC rainfall cores is expected to be more
discernible compared to DCCs with significant large-scale forcing, given the limited large-scale ascent and minimal convection
organization in such cases.

Throughout the DCC days, the Houston-Galveston region experienced diverse aerosol number concentrations. As shown in
Figure 7, the distribution of aerosol number concentrations spans a considerable range with a prominent peak at smaller number

concentration bins. The mean values of these SS-determined distributions are significantly different according to results from
a t-test, except for $N_{ccn1}$ and $N_{ccn08}$. More specifically, this environment exhibits mean total aerosol number concentrations
of 7,332 $cm^{-3}$ for $N_{cn}$ and 10,683 $cm^{-3}$ for $N_{ufp}$ during the study period (Figures 7g, h), showing high pollution levels. The
most polluted instances occurred in mid-July (e.g., July 12, 13) and mid-August (e.g., August 10, 11, 17), exceeding the 95th
percentile values of the distributions shown in Figure 7.

In addition, the Houston-Galveston region is found to have a unique combination of different aerosol species during the
summer months (Figure S5, in the supporting information), according to the aerosol mass concentration measurements at the
M1 site. The predominant aerosol type measured is total organics, constituting 49% (2.24 $\mu gm^{-3}$) of the total aerosol mass
concentration, followed by sulfate at 34% (1.54 $\mu gm^{-3}$), ammonium at 13% (0.61 $\mu gm^{-3}$), nitrate at 3% (0.14 $\mu gm^{-3}$), and
chloride at $< 1\%$ (0.03 $\mu gm^{-3}$). This broad spectrum of aerosol species and their mass concentration is indicative of various

emission sources, including both anthropogenic (e.g., from city, ships, refineries; Rivera et al., 2010; Wallace et al., 2018) and



natural emissions (e.g., from agricultural activities, vegetation; Bean et al., 2016; Yoon et al., 2021) from nearby and/or distant locations.

## 4.2 Average aerosol causal effects on DCC ETH

Figure 8 illustrates the estimated average causal effect of aerosol number concentration on 30-dBZ ETH and 15-dBZ ETH,
for all scenarios and varying distances (20 to 50 km) from the M1 site. The valid scenarios are indicated by the white hatch lines, which are determined in Section 3.2. The confounding meteorological variables are calculated using the most unstable parcel in this figure, and the post-sounding aerosol averaging method is used. The findings reveal a positive average causal effect for $N_{cn}$ and $N_{ufp}$, when all aerosol particles are activated in convective updrafts. It implies that higher aerosol number concentration values correspond to an increase in 30-dBZ ETH within DCCs, thereby suggesting a stronger convective updraft
in polluted conditions compared to that in clean conditions. However, the expected causal effects of aerosols on the 30-dBZ ETH show only moderate variations when using different exposure variables in the causal model, ranging between 0.6 km to 2.2 km. The mean aerosol causal effect across these valid scenarios is 1.0 km or 13% of the average 30-dBZ ETH. We observe similar results when using 15-dBZ ETH as the outcome variable and when using different air parcels for calculating confounding variables, as illustrated in Table 3.

Interestingly, when conducting causal analysis on the invalid scenarios, the estimated average aerosol causal effects are mostly negative (Figure 8). This highlights the importance of evaluating the causal model or any prediction model used in similar studies, as without this evaluation, contradictory results may be obtained.

In a separate test, we ran the causal model without any confounders, and we found that the estimated mean aerosol causal effects on 30-dBZ ETH increased to 1.4 km, which is 0.4 km larger than when including two confounders. These results
highlight the importance of considering confounders while quantifying aerosol impacts on convective properties.

Note that the observational findings presented in this study do not unequivocally lend support to or negate the previously proposed warm-phase invigoration pathway. The role of in-cloud SS is vital in determining the occurrence of warm-phase invigoration within DCCs (e.g., Romps et al., 2023). Unfortunately, direct *in situ* measurements of SS within convective updrafts remain unavailable, despite estimates using aircraft measurements for limited climate and vertical velocity regimes
(e.g., Politovich and Cooper, 1988; Pinsky and Khain, 2002; Korolev and Mazin, 2003; Prabha et al., 2011; Romps et al., 2023). The aerosol invigoration effect in our study is substantiated based on the assumption that in-cloud SS exceeds a certain threshold to activate all aerosol particles. In essence, the results do not directly support warm-phase invigoration unless in-cloud SS is measured or estimated in line with our assumptions.

Similarly, the presented causal effects do not conclusively confirm or reject the possibility of other hypothesized aerosol
invigoration mechanisms (e.g., cold-phase, entrainment-humidity invigoration). As shown in Figure 6b, a substantial portion of the 30-dBZ ETH associated with the studied rainfall cores extends beyond 5 km. Consequently, the observed positive causal effects of aerosols under specific conditions suggest potential evidence of cold-phase invigoration or partitioning between warm- and cold-phase invigoration. However, to fully support these invigoration mechanisms, we need to further assess the relative importance of additional latent heat release and hydrometeor loading (e.g., Igel and van den Heever, 2021). It requires



crucial supporting measurements of hydrometeor and latent heating profiles in the convective updraft region, which were not available during the majority of the TRACER IOP. Moreover, while entrainment is found to alter aerosol-DCC interactions (e.g., Peters et al., 2023), the absence of vital, direct measurements of convective vertical velocity, presents a challenge in evaluating the significance of this process.

In summary, we demonstrate a causal link between aerosol number concentrations and ETH using various observational
data sets through a novel application of the g-computation model. However, to gain a comprehensive understanding of the plausible pathways driving aerosol-induced effects on ETH necessitates advanced instrumentation and specific field campaign designation, which are capable of capturing SS levels, vertical velocity within updrafts and understanding the intricate dynamics and microphysical processes occurring within DCCs.

### 4.3 Impacts of the sea-breeze circulations on aerosol causal effects

The ARM M1 site is located in close proximity to Galveston Bay (6 km) and the Gulf of Mexico (50 km), frequently experiencing Bay- and Gulf-breeze circulations (simplified as sea-breeze circulations in the following text) during the summer months (Wang et al., 2024). Despite focusing on cases within the anticyclonic regime to exclude large-scale ascent contributions to the development of DCCs, sea-breeze fronts can still act as meso-scale forcing mechanisms, inducing upward motions within the boundary layer and influencing aerosol-DCC interactions.

Our recent study (Wang et al., 2024) indicates that at least 44% of the DCC rainfall cores analyzed here are associated with days that these circulations are present. In that study, we identified sea-breeze circulation days based on observations from NEXRAD, Geostationary Operational Environmental Satellites (GOES), and ARM surface meteorology data (e.g., wind fields, water vapor mixing ratio). Specifically, 64 sea-breeze circulation cases were determined during the TRACER IOP. As shown in Table 1, 38 rainfall cores, with a sounding launch within 6 hours prior to rainfall initiation, were tracked during these
days within 50 km of the ARM M1 site in this study.

We applied the causal framework to DCCs that are associated with sea-breeze circulations, maintaining the same confounding variables since they showed moderate correlations with both outcome and exposure variables ($R$ values ranging from 0.4 to 0.5). Figure 9 illustrates the causal effect on 30 dBZ ETH as the outcome variable. The mean causal effect observed is 1.4 km, which is higher than estimates for scenarios including all cases (Table 3). This increase could be due to the potential exclusion
of confounding variables that are not major contributors to non-sea-breeze cases. One important variable could be boundary layer updrafts, which consistently increase at the leading edge of sea-breeze fronts as observed from the Doppler Lidar measurements (Wang et al., 2024). Since this observation is only available at the ARM M1 site and not for each tracked rainfall core, it is challenging to include this confounding variable in the causal model. The exclusion of this confounding variable may lead to an overestimation of the causal effects of aerosols on ETH as discussed in the previous section.

Interestingly, we found more valid causal models (19) for the sea-breeze cases compared to scenarios including all cases. This suggests that the aerosol influence is a robust signal here, even though the extension of the ETH is not more than 15%. This robustness may be due to the coherent separation of clean versus polluted cases when using different exposure variables. This is supported by the observations that the DCC environment is much cleaner after the passage of sea-breeze fronts.



### 4.4 Average aerosol causal effects on precipitation intensity and area

In this subsection, we extend our causal framework to estimate the impacts of aerosols on precipitation intensity and area. Precipitation intensity is assessed using the maximum 2-km radar reflectivity, while precipitation area is evaluated based on the maximum area with 2-km $Z > 30$ dBZ of the tracked precipitation core throughout the cell life cycle. All steps in the causal framework remain the same for these applications, except the outcome variable is either maximum radar reflectivity or precipitation core area instead of ETH. The confounding variable considered in this analysis is only CAPE, as it is the only 530 one that shows a correlation coefficient higher than 0.3 with both outcome and exposure variables.

Figure 10 presents the causal effects on the core area for different potential exposure variables. Only one causal model is valid, which corresponds to the scenario with DCCs identified within 30 km of the M1 site using $N_{ccn}$ measured at SS of 0.8%. This finding implies that, only on rare occasions, aerosol number concentration impacts the precipitation core area expansion by approximately 39 $km^2$. Given the fact that this area expansion is only observed in limited scenarios, it is less conclusive 535 compared to the effects of aerosols on DCC ETH.

Regarding the causal effects of aerosols on precipitation intensity, Figure 11 shows ten effective models, significantly more than those considering core area as the outcome variable. Although the mean causal effect across all valid scenarios is positive, the magnitude is around 2 dBZ, which falls within the uncertainty range of the NEXRAD radar (3 dBZ; Gou, 2003; Ryzhkov et al., 2005). Therefore, we cannot conclusively determine that aerosol loading results in heavier precipitation for the DCC 540 cases evaluated in this study.

### 4.5 Sensitivity of the causal effect estimation

We explore the robustness of aerosol causal effect estimates by examining various factors that could influence the calculations. These factors include the data averaging period for aerosol measurements and the time gap between environmental measurements and DCC rainfall initiation.

When using the prior-rain method for the aerosol averaging process, as shown in Figures 3b, d, we observe that the effective aerosol properties (exposure variables) remain consistent with those obtained using the post-sounding method (Figures 3a, c), involving $N_{cn}$ and $N_{ufp}$. The mean aerosol effect on 30-dBZ ETH/15-dBZ ETH is 1.1 km/1.0 km, which aligns closely with the results obtained using the post-sounding aerosol averaging method (Table 3). These findings suggest that the causal model results have minimal sensitivity to the data averaging period for the measured aerosol properties used in this study.

Regarding the influence of the time gap between measurements of DCC and environmental properties on the estimation of the aerosol causal effect, we exclude the cases when the nearest soundings were launched more than 4 hours before the initiation of DCC rainfall cores (Table 3). As shown in Table 3, the mean aerosol effect on 30-dBZ ETH/15-dBZ ETH across all valid scenarios is 1.2 km/1.2 km, only slightly higher than using 6-hour soundings.

The shorter the time difference, in theory, the more accurately the sounding measurement should represent the environment 555 in which the DCCs are embedded. Therefore, these results reinforce the conclusion from previous sections, suggesting that aerosol invigoration is, for the most part, constrained, and requires all aerosol particles to be activated in convective updrafts if



it is to be effective. However, the number of samples is reduced by approximately 20% when limiting our analysis to 4-hour soundings. Additionally, the percentage of cases heavily influenced by sea-breeze circulations also changes. These changes could all potentially impact the casual model results.

In summary, the assessment of aerosol causal effects appears independent of the timing of environmental measurements relative to the initiation of DCCs and the accuracy with which these measurements reflect the air ingested into the DCC updraft cores. Nonetheless, the collective findings indicate a restricted impact of aerosols on DCCs across all sensitivity tests conducted in the Houston-Galveston region under anticyclonic regimes.

### 4.6   Potential uncertainties in causal analysis

The g-computation model is a flexible and powerful technique, but its application to observational data necessitates careful consideration of assumptions and potential sources of bias. One major challenge when estimating the causal effect of an exposure is controlling for unobserved or unknown confounders (e.g., Barrowman et al., 2019; Hjellvik et al., 2019). The presence of unobserved/unknown confounders may cause the observed data distribution to be compatible with many contradictory causal explanations.

In this study, we have accounted for important confounders that could influence the aerosol-DCC interactions according to previous studies and also our evaluations, but there may still be some confounders that we did not observe or discover that could impact our results. For example, Peters et al. (2023) discovered that entrainment rate influences whether aerosols have an impact on DCCs. Additionally, the size of the updraft core in the boundary layer prior to the cloud formation is identified as a significant factor influencing the intensity of the subsequently developed DCCs (e.g., Morrison, 2017; Mulholland et al., 2021;

Takahashi et al., 2023). However, direct measurements of these quantities were not available during TRACER and most other field campaigns aimed at observing the characteristics of deep convection. The lack of confounders in the causal model may possibly cause an overestimation of the aerosol causal effects. Nonetheless, even though all the confounders are observed, to balance the number of samples and the number of confounders in the causal model, these confounders may not all be included in the model (as discussed in Section 3.3).

Recently, several potential solutions have emerged that show promising results in overcoming this challenge (D'Amour, 2019; Peterson et al., 2023). For example, Liu et al. (2020) controlled for unobserved confounders in a novel manner by using double binary confounders that satisfy a nonlinear condition on the exposure. Various simulations show better estimation performance compared to the current approach. Such simulations will be considered in our future studies.

### 5   Conclusions

This study introduces a novel application of the g-computation causal inference model to explore the causal effects of aerosols on the rainfall core properties of DCCs, aiming to provide evidence of aerosol invigoration or enervation. Leveraging the extensive observational dataset collected during the TRACER IOP (Jun. - Sep.) in the Houston-Galveston region, characterized by a diverse aerosol environment, we focus on examining isolated DCCs observed during this period in the anticyclonic regime.



To identify suitable DCCs for investigation, we establish an interpretable framework including a three-step process.

First, we exclude synoptic-scale system-driven cases by applying a regime classification of synoptic weather patterns using the SOM method. This step allows us to focus on locally driven cases under anticyclonic regimes, which are found to be more conducive to aerosol interactions in previous studies. The selected period is characterized by low shear, limited large-scale uplift, and moderate humidity conditions, favoring predominantly isolated DCCs driven by local factors.

Second, we track DCC cases initiated within a certain distance from the M1 site using a Lagrangian framework based on NEXRAD data. This tracking process helps determine the properties of the DCC rainfall cores, which are identified as small in size ($74 \ km^2$ on average), slow propagating ($5 \ ms^{-1}$ on average), and short-lived (51 min on average), with predominantly afternoon initiation likely influenced by surface heating flux and/or sea-breeze circulations. In particular, 44% of the DCC cells tracked occurred on sea-breeze days.

Finally, we use the g-computation model to assess the causal effect of aerosols on identified DCCs. Before implementing the model, we categorize observed variables into three groups: exposure, confounder, and outcome variables. The outcome variables, representing updraft strength, are 30-dBZ ETH or 15-dBZ ETH. From eight aerosol parameters ($N_{ccn}$ at six SSs, $N_{cn}$, and $N_{ufp}$), we select exposure variables by evaluating the performance of SLR models, where the relationships between these variables and the outcome variables are fitted. Only a small fraction (16%) of the SLR models are valid, indicating that, in the majority of cases, aerosol loading is not associated with the evolution of DCC ETH, suggesting insufficient effects of aerosols on DCC updraft velocity in these situations. For confounders, we identify two observed convective indices that covary with aerosol and DCC, (CAPE and $ELR_3$ for 30-dBZ ETH, CAPE and LWS for 15-dBZ ETH) and need to be considered when estimating aerosol effects. In the g-computation model, we initially fit a Q-model (MLR model in our case), where the outcome is modeled as a function of the exposure and relevant confounders. Subsequently, the fitted MLR model is employed to predict counterfactual outcomes for each observation under each exposure scenario (clean or polluted). Finally, the average causal effect is calculated by taking the difference between the average counterfactual outcomes under the clean and polluted conditions.

The major findings include:

1. After accounting for confounders, we observed a positive average causal effect of aerosols on DCC ETH, ranging from 0.7 to 1.2 km. This result suggests that in these particular scenarios, a stronger convective updraft may be expected in a more polluted condition compared to those in a clean condition, everything else being equal or similar. This influence is optimal when the SS levels in the convective updates support the activation of all aerosol particles injected.

2. When assessing the impacts of sea-breeze circulations on aerosol-DCC interactions, we found a slightly higher impact of aerosol number concentration on 30-dBZ ETH (0.4 km deepening) compared to the all-case scenario. This discrepancy could be due to the absence of major confounding variables (e.g., boundary layer dynamics) considered in the causal model for this scenario.

3. We also apply the causal framework to investigate the impact of aerosol loading on precipitation core area. There is only one model that shows valid results, indicating an expansion of cell area in the polluted cases. However, the robustness





of this result needs further assessment. Moreover, regarding the influence of aerosols on maximum rainfall intensity, the observed effects fall within the range of radar measurement uncertainty.

4. The sensitivity analysis reveals minimal dependency on the choice of the proxy for updraft intensity, the temporal and spatial gaps between measurements of aerosol and DCC properties, the aerosol averaging period, and the types of originating air parcels used in calculating CAPE. In other words, these tests all show comparable causal effects of aerosols on 30-dBZ ETH.

Nevertheless, this study pioneers the use of a causal machine learning model to assess the effects of aerosols on DCC prop-
erties, marking a significant advance in the study of aerosol-convection interactions using observations. This novel application also documents the first step in quantitatively unraveling the complexities of aerosol-meteorological co-variability on aerosol-convection interactions. Furthermore, this causal framework holds broad implications for other topics and fields, offering a powerful new tool for addressing scientific questions.

*Code and data availability.*

1. ARM data can be downloaded from https://adc.arm.gov/discovery/#/

2. NEXRAD data is accessible at https://registry.opendata.aws/noaa-nexrad/

3. TINT package: https://github.com/openradar/TINT

*Author contributions.* DW and MJ designed the study. DW conducted the analysis and wrote the manuscript. RK and TZ provided guidance on running the causal model. TS provided the aerosol dataset. SVDH, SG, and MJ reviewed the manuscript.

*Competing interests.* The authors have no conflicts of interest to declare.

*Acknowledgements.* This paper has been authored by employees of Brookhaven Science Associates, LLC, under Contract DE-SC0012704 with the U.S. Department of Energy (DOE). We would like to acknowledge the DOE Early Career Research Program for the funding support. We also would like to acknowledge support from the Atmospheric System Research (ASR) program, the Atmospheric Radiation Measurement (ARM) user facility, and the ARM TRACER operation and science teams. SV is supported by the DOE under DE-SC0021160.
SG is supported by Argonne National Laboratory under U.S. DOE contract DE-AC02-06CH11357 and the ARM User Facility, funded by the Office of Biological and Environmental Research in the U.S DOE Office of Science. We thank Maria Zawadowicz at Brookhaven National Lab for providing the calculated CCN number concentrations.





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



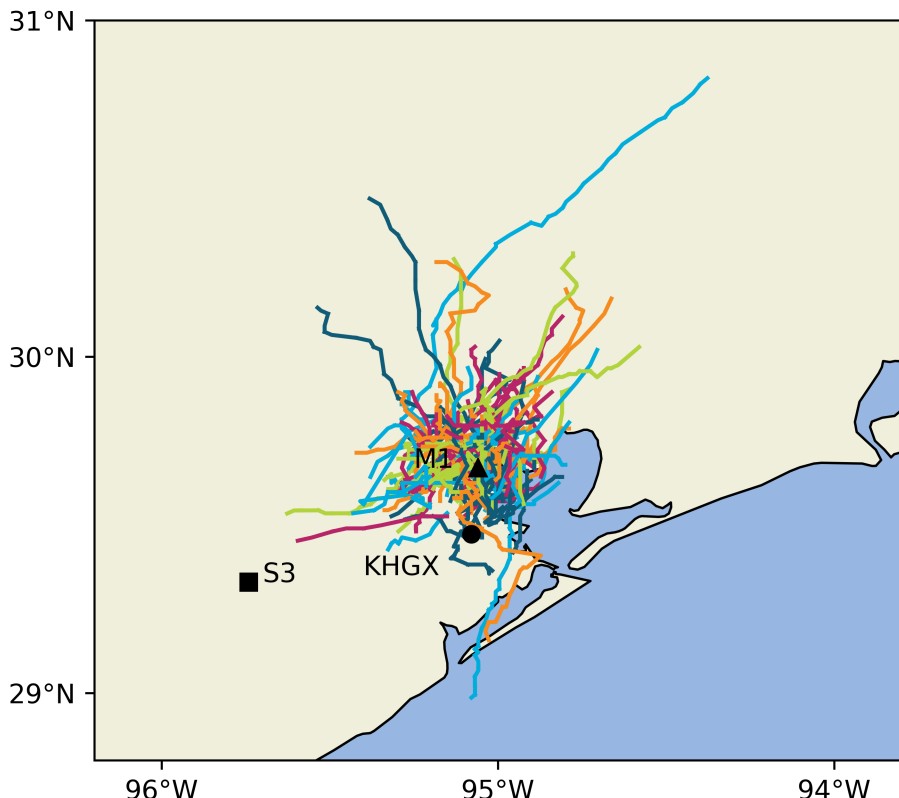

**Figure 1.** Maps of locations of DCC rainfall core tracks identified for cells initiated within a radius of 20 km from the M1 site. The black triangle (M1) indicates the location of the ARM mean site at La Porte, TX, the black circle indicates the location of the NEXRAD KHGX radar, and the black square (S3) indicates the location of the ancillary site at Guy, TX. The colors have no specific meaning other than to identify individual tracks.



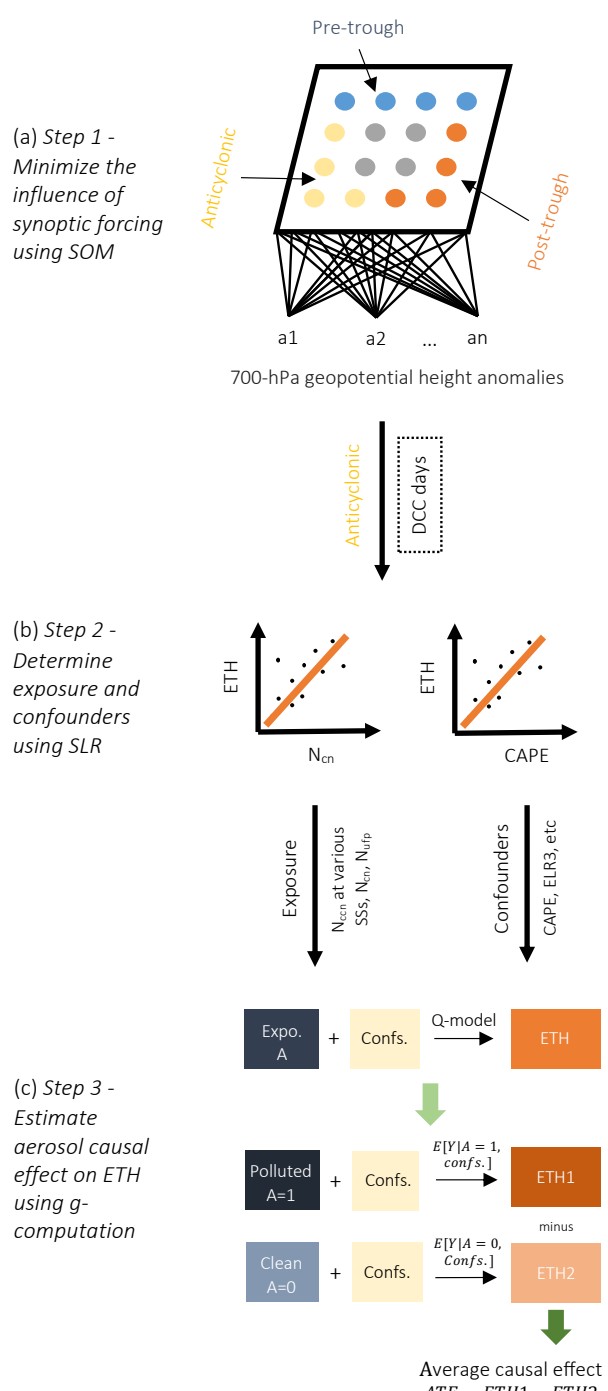

**Figure 2.** Flow chart of the causal model framework used to estimate the causal effect of aerosols on DCC ETH.





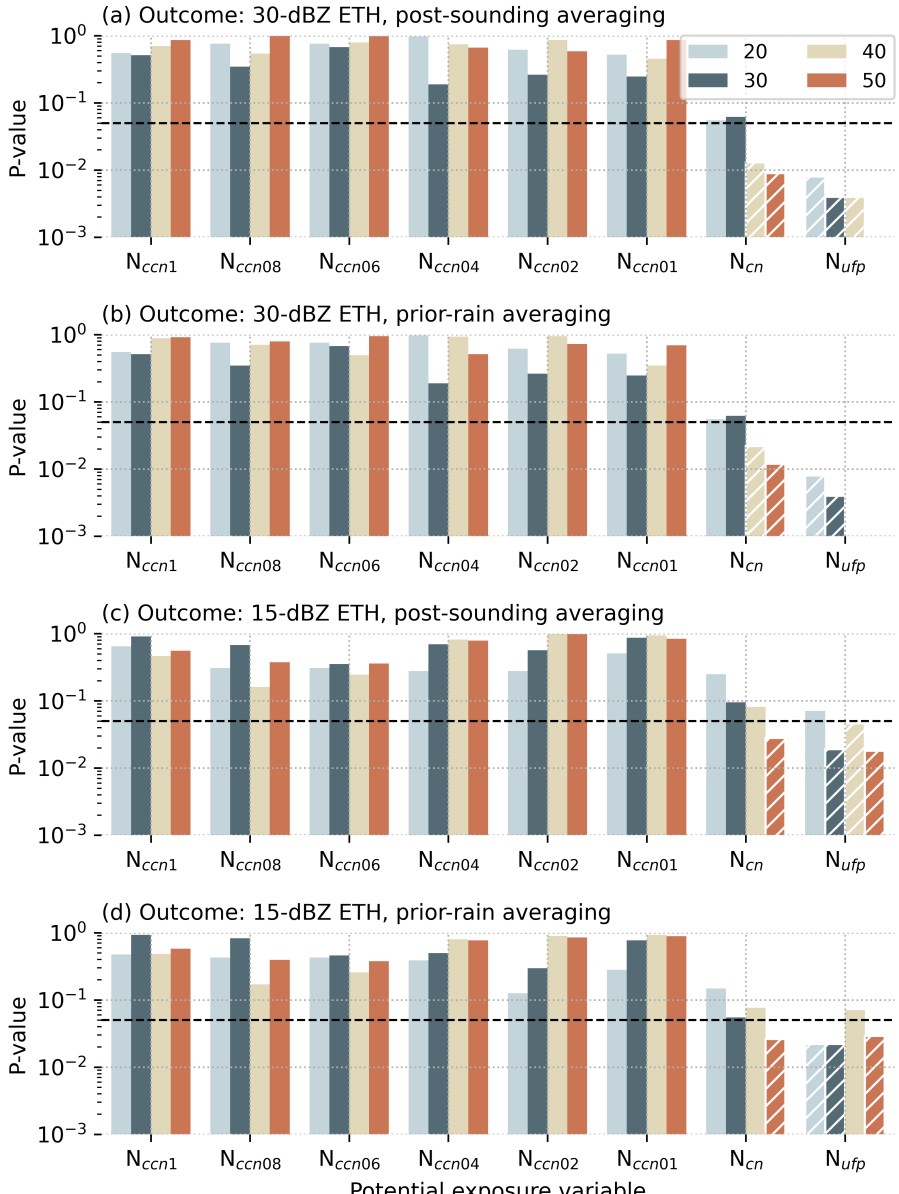

**Figure 3.** Simple linear regression model $P$-value for each aerosol number concentration as a predictor (potential exposure variable) respectively for different aerosol averaging periods. Different colors represent different maximum distances between aerosol and DCC measurements (km in radius from the M1 site). The horizontal line indicates $P = 0.05$ and the white hatch lines indicate valid models ($P < 0.05$). Note that for some models, the $P$-value is zero which is not visible on the plot.





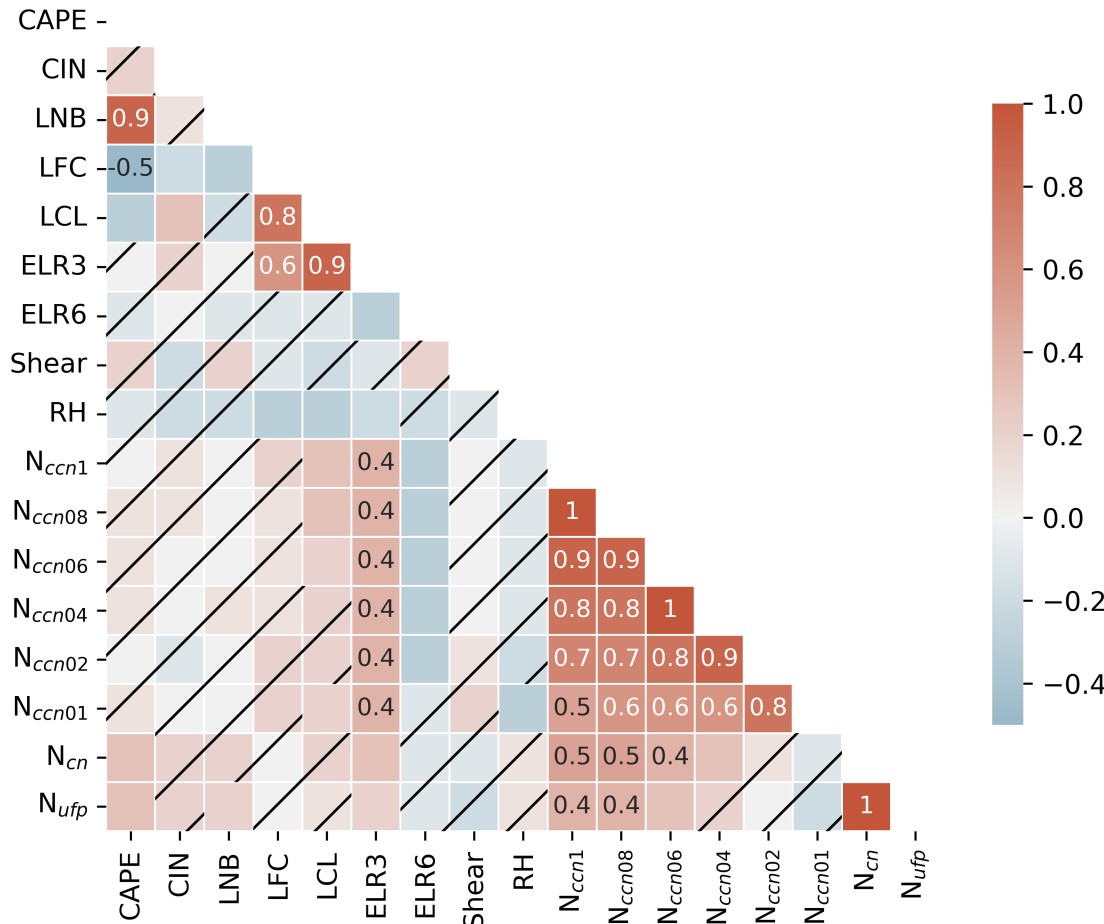

**Figure 4.** Correlation matrix between the meteorological variables and aerosol number concentrations for DCC cases identified within a radius of 50 km from the M1 site. The correlation matrix is a table showing Pearson $R$-values between sets of variables. The meteorological variables are calculated using ARM soundings when assuming the most-unstable parcel would rise to form a convection. The black hatch lines indicate non-significant $R$-values.




**Figure 5.** Histograms with density kernel estimation (solid lines) of meteorological variables from the ARM soundings launched prior to DCC cases identified within a radius of 50 km from the M1 site. The bin size is defined by the difference between the maximum and minimum values of each variable divided by the number of bins, which is fixed at 10 for each subplot.





**Figure 6.** Histograms with density kernel estimation (solid lines) of the maximum 2-km radar reflectivity, 30-dBZ ETH, and 30-dBZ rainfall core area along with initiation time, mean propagation speed, lifetime for each DCC rainfall core identified within a radius of 50 km from the M1 site. The binwidth is set to 2 dBZ for (a), 1 km for (b), 20 km$^2$ for (c), 2 hrs for (d), 0.5 m/s for (e), and 10 min for (f).





**Figure 7.** Histograms with density kernel estimation (solid lines) of CCN number concentrations measured at different SS levels and total aerosol number concentrations for DCC cases identified within a radius of 50 km from the M1 site. The binwidth is set to 500 $cm^{-3}$ for (a)-(d), 200 $cm^{-3}$ for (e), 20 $cm^{-3}$ for (f), 1,000 $cm^{-3}$ for (g), and 2,000 $cm^{-3}$ for (h).



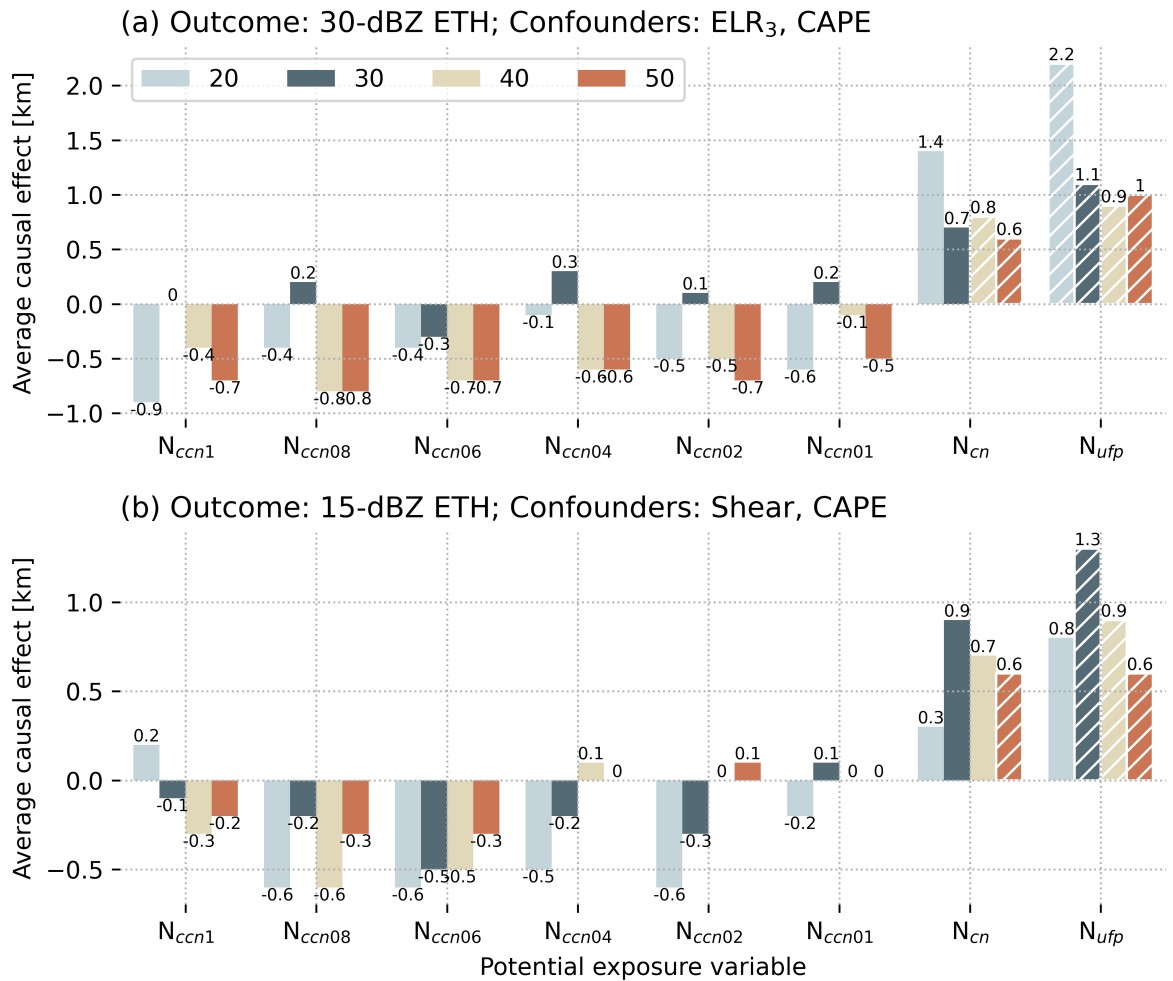

**Figure 8.** Average causal effects on (a) 30-dBZ ETH and (b) 15-dBZ ETH estimated for each potential exposure variable after controlling for confounders. Different colors represent different maximum distances between measurements of environmental variables and DCC properties. The meteorological variables are calculated using ARM soundings (6-hr) when assuming the most-unstable parcel would rise to form a convection. The white hatch lines indicate valid results.





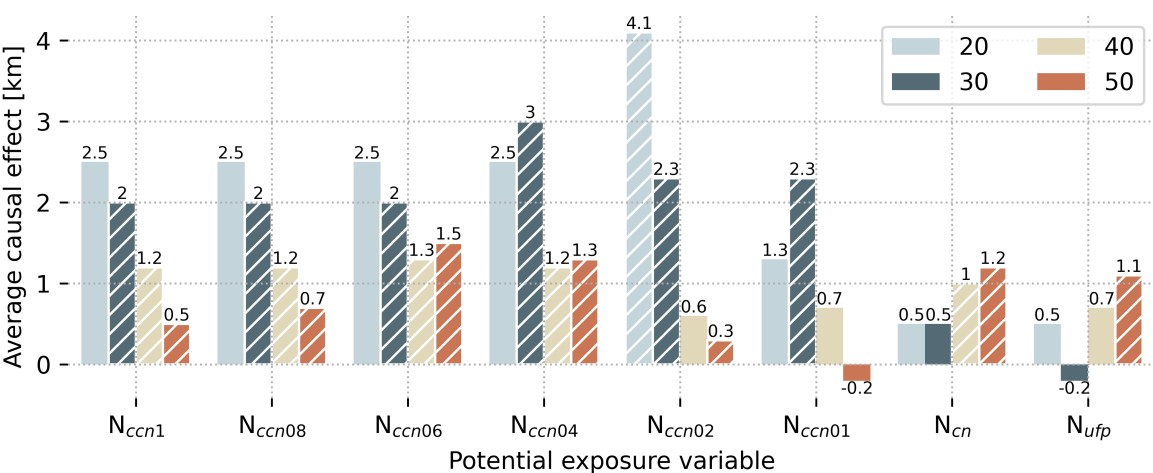

**Figure 9.** Average causal effects on 30-dBZ ETH estimated for each potential exposure variable after controlling for confounders (ELR$_3$ and CAPE) for DCCs identified during sea-breeze days only. The post-sounding aerosol averaging period is considered. Different colors represent different maximum distances between measurements of environments and DCCs. The meteorological variables are calculated using ARM soundings (6-hr) when assuming the most-unstable parcel would rise to form a convection. The white hatch lines indicate valid results.





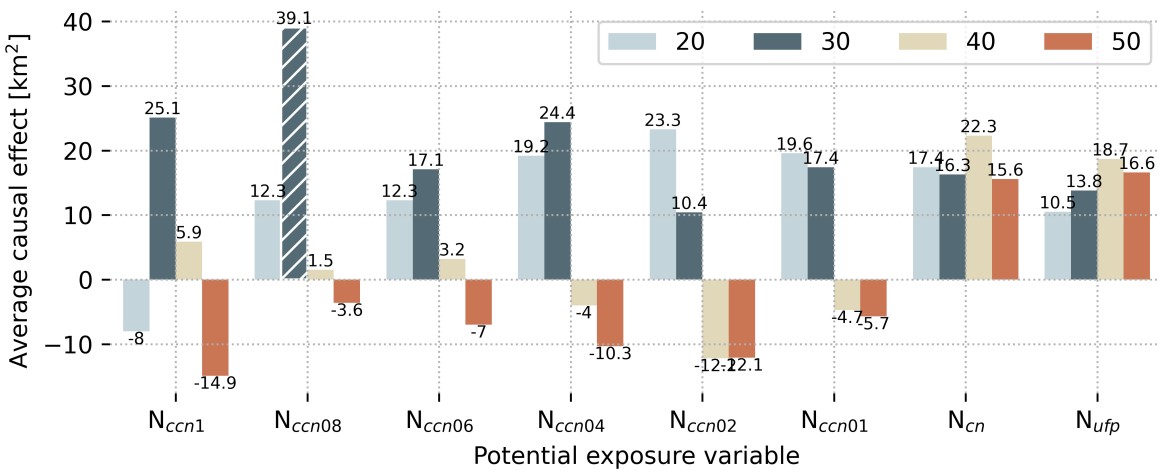

**Figure 10.** Same as Figure 8a but for cell area as outcome variable and CAPE as confounding variable.



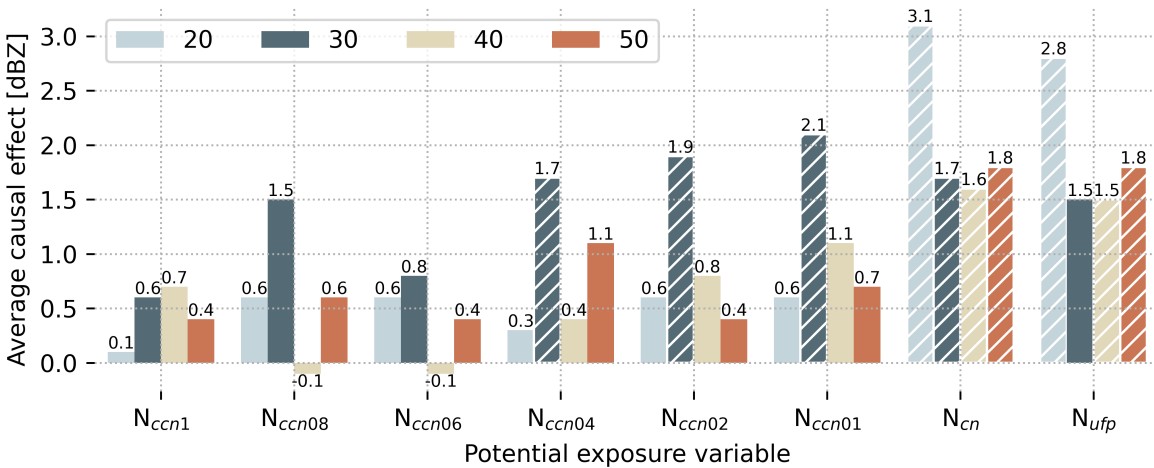

**Figure 11.** Same as Figure 8a but using maximum reflectivity as outcome variable and CAPE as confounding variable.





**Table 1.** Number of DCC cases tracked in 2022 from June to September when considering different radii to the M1 site and under different scenarios.

| Distance to M1 | 6-hr soundings | 4-hr soundings | Sea-breeze days, 6-hr soundings |
|:---:|:---:|:---:|:---:|
| 20 km | 43 | 29 | 12 |
| 30 km | 61 | 46 | 22 |
| 40 km | 70 | 54 | 29 |
| 50 km | 86 | 70 | 38 |





**Table 2.** Pearson correlation coefficients ($R$-values) between convective indices and DCC ETH. The most-unstable parcel is used in the calculations of the convective indices. DCCs were identified within different distances, ranging from 20 to 50 km, from the ARM M1 site. Only the $R$-values that pass the significance tests are included.

| Distance to M1 | LNB | CAPE | LCL | LFC | ELR$_3$ | ELR$_6$ | LWS | RH |
|---|---|---|---|---|---|---|---|---|
| *Outcome variable: 30-dBZ ETH* | | | | | | | | |
| 20 km | × | × | × | × | × | × | × | × |
| 30 km | × | × | × | × | 0.3 | × | × | × |
| 40 km | × | 0.3 | 0.2 | × | 0.3 | × | × | × |
| 50 km | 0.2 | 0.2 | 0.3 | × | 0.4 | × | × | × |
| *Outcome variable: 15-dBZ ETH* | | | | | | | | |
| 20 km | × | × | × | × | × | × | -0.3 | × |
| 30 km | × | × | × | × | × | × | × | × |
| 40 km | × | × | × | × | × | × | × | × |
| 50 km | 0.2 | 0.3 | × | × | × | × | × | × |



**Table 3.** Average causal effects on ETH [km] using different confounders and outcome variables under different scenarios.

| Confounders | $CAPE_{mu}$, $ELR_3$ | $CAPE_{sfc}$, $ELR_3$ | $CAPE_{mix}$, $ELR_3$ |
|---|---|---|---|
| *6-hr soundings, post-sounding averaging* | | | |
| 30-dBZ ETH | 1.0 | 1.0 | 1.0 |
| 15-dBZ ETH | 0.8 | 0.7 | 0.8 |
| *6-hr soundings, prior-rain averaging* | | | |
| 30-dBZ ETH | 1.1 | 1.1 | 1.1 |
| 15-dBZ ETH | 1.4 | 0.8 | 0.8 |
| *4-hr soundings, post-sounding averaging* | | | |
| 30-dBZ ETH | 1.2 | 1.2 | 1.2 |
| 15-dBZ ETH | 1.2 | 1.2 | 1.3 |
| *4-hr soundings, prior-rain averaging* | | | |
| 30-dBZ ETH | 1.2 | 1.2 | 1.2 |
| 15-dBZ ETH | 1.1 | 1.1 | 1.2 |