# Peer review of "Aerosol Impacts on Isolated Deep Convection: Findings from TRACER"

_EGUsphere, 2024_

## Author Comment (AC1)

We sincerely thank the reviewers for their thoughtful comments, insightful questions, and constructive suggestions. Their feedback has greatly helped us clarify our ideas, strengthen our arguments, and improve the overall quality of the manuscript. Our responses are in blue text.

Major Comments.

1) Readability: The new statistical approach is meticulously written, but it is often hard to read due to various statistical jargon that is unfamiliar to atmospheric scientists like me. Can you clearly define these terms at the beginning? For example, define it like this in the table.

Confounder: a variable that affects both the dependent and independent variables in a study, causing an association that may not be accurate. (parameters include ….)

Exposures: Any factor that may be associated with an outcome of interest. (parameters include ….)

Probably these terms are common in epidemiology, but not in atmospheric science.

We thank the reviewer for this suggestion. It would indeed be helpful for the reader to be acquainted with these terms earlier in the manuscript. We have added a table and a few sentences about these variables to the introduction where the g-computation model is first introduced. This information is provided to the reader again in Section 3 where the g-computation analysis begins.

We added these sentences to Line 99 in the original manuscript: "In general, g-computation requires the identification of three types of variables for causal analysis: the exposure variable, the outcome variable, and the confounder variable(s). These variables are described in Table 1 and further explained in Section 3."

**Table 1.** Explanations of each term in the g-computation model with examples.

| Terms | Explanations | Examples |
|---|---|---|
| Exposure variable / Independent variable | It is a variable whose causal effect on another variable (outcome) is being investigated. It represents the condition being manipulated or analyzed in hypothetical scenarios. | Aerosol number concentration, CCN number concentration |
| Outcome variable / Dependent variable | It is the variable of interest for which we aim to estimate the causal effect of an exposure. By applying g-computation, potential outcomes under varying exposure levels can be simulated, allowing for the assessment of differences between exposure scenarios. | Convective cloud 30-dBZ echo top height |

| Confounder / Confounding variable | They are variables, other than the one being studied (the exposure), that are associated with both the outcome and the exposure. They can distort or mask the true effect of the exposure on the outcome, leading to inaccurate conclusions about the relationship between the two. | Convective Available Potential Energy, Environmental Lapse Rate |
|---|---|---|

2) New and traditional approach: At the end of the manuscript, the authors mentioned quite significant statements "Nevertheless, this study pioneers the use……….. scientific questions". To be honest, I still wonder why this new method is so novel compared to the previous old approach because there's no comparison between the new and traditional statistical approaches. For example, here is one of the earliest aerosol-deep convection manuscripts.

Lin, J. C., Matsui, T., Pielke, R. A., & Kummerow, C. (2006). Effects of biomass-burning-derived aerosols on precipitation and clouds in the Amazon Basin: A satellite-based empirical study. Journal of Geophysical Research: Atmospheres, 111(D19). https://doi.org/10.1029/2005JD006884

In this paper, DCC properties (precipitation, cloud top height, and cloud fraction) are related to aerosol optical depth for a given meteorological parameter (cloud work function in that study). Can you compare your novel approach with this traditional approach (simple statistics stratified by meteorological parameters)? Do you think the old approach leads to significant biases in understanding the aerosol-DCC relationship? Can you prove or briefly explain?

We first thank the reviewer for providing the reference. (We added it to the introduction.)

In Lin et al. (2006), they relied on bivariate correlations, which do not account for basic confounding effects. In contrast, our method extends previous capabilities and attempts to control for confounding using known or potentially confounders.

We are not denying the fact that linear regression or correlation can be used for causal inference, but only under ideal circumstances where individual values are *randomly assigned* to groups. This condition, however, is not applicable to our observational study or similar types of studies in atmospheric science. Fundamentally, whether linear regression can infer causal relationship depends on how the data was collected. See the first few paragraphs of the introduction in Chatton et al. (2020) for more information.

In the case of aerosol-convection interactions, it is *impossible* to erase the background aerosol state and randomly inject specific amounts of cloud condensation nuclei (CCN) into naturally formed convective clouds with current technologies. The CCN concentration at a particular location on a

given day can be a result of other factors, such as humidity, wind direction, and/or pre-existing convection. These hidden factors (confounders) could themselves be the true causes of changes in convective intensity. As long as CCN concentrations cannot be randomly assigned, bivariable correlation coefficients cannot accurately infer causal effects. In some cases, bivariate correlations can lead to more bias compared with g-computation results as discussed in Snowden et al. (2011).

In our method, we control the aerosol state through a "forced" experiment, which, though less ideal than a fully randomized experiment, involves adjusting certain variables while others are held constant or randomized to minimize their confounding effects. In our case, we forced the aerosol number concentration to be 1 as a polluted condition and 0 as a clean condition. Our identified confounders were kept constant in these two scenarios.

Additionally, g-computation offers a more flexible framework. While we currently use linear regression as our Q-model in the first step, it can support more complex methods, such as machine learning regression, to accurately capture nonlinear relationships. In contrast, correlation analysis is limited to detecting linear relationships.

3) Potential biases in radar-based approach: Authors use threshold NEXRAD radar parameters to define DCC. However, if DCC has a much smaller amount of raindrops due to a large number of background aerosols, this cell may not be counted as DCC due to larger concentrations of small-size droplets, which won't increase S-band reflectivity. Alternatively, if you use cloud optical depths and top height, the DCC sampling can include such cells. This is a NEXRAD-based cell tracking approach, so you cannot change your approach. However, it is important to discuss potential sampling biases using the NEXRAD radar.

We thank the reviewer for pointing this out. We agree that using fixed thresholds on radar reflectivity for tracking cells may introduce potential uncertainties in sample selection.

To address this, we have added the following sentences to Section 2.1 in the manuscript: "Note that using fixed thresholds may potentially influence the selection of DCCs investigated in the study, particularly in conditions where DCCs contain fewer raindrops due to the presence of a large number of background aerosols."

Minor Comments.

Line 87: Please remove parenthesis "(either invi….. )".

Agreed

Line 120-121: "exclude the presence of shallow convection" sounds like removing the sampling during shallow stages. So I suggest just re-write as "exclude the shallow convection cells".

Agreed

Line 179: Please define the threshold of diameters of "ultrafine aerosols".

Done

Line 274: "buoyancy-driven DCCs". Well, all DCCs are driven by buoyancy over the flat terrain. So you may re-write this as "locally driven DCCs".

Agreed

Line 294: "30-dBZ ETH/15-dBZ ETH" should be "30-dBZ ETH and15-dBZ ETH".

Done

Line 303-306: We won't be able to measure supersaturation directly within the convective storms. However, you can infer the required supersaturation in order to activate all aerosols (including ultrafine). For this case, can you describe roughly how much supersaturation is required to support your argument?

The exact supersaturation required to activate all aerosol particles in a particular environment is challenging to estimate without appropriate instrumentation. The actual supersaturation values may depend on meteorological conditions, including atmospheric instability, moisture content, and updraft strength. A SS value of 1% does not yield a statistically meaningful effect of $N_{ccn}$ on DCC ETH in our study (Figure 8 in the manuscript). We hypothesize that a higher SS (> 1%) may be necessary to activate more particles and effectively influence DCCs in the Houston region. However, we have refrained from adding this discussion of hypothetical SS values needed to activate additional aerosols within the manuscript. This is because the values remain speculative and are not based on actual observations of SS within the convective clouds.

Fig. 4: Why is there no correlation between thermodynamics and $N_{ccn}$? It seems to be more important?

The black hatch lines indicate non-significant R-values on Figure 4, meaning these values are not statistically significant. Basically, there are no significant correlations between most of the environmental variables and $N_{ccn}$ in Houston.

Line 547: "30-dBZ ETH/15-dBZ ETH is 1.1 km/1.0 km," should be "30-dBZ ETH and15-dBZ ETH is 1.1 km and 1.0 km, respectively."

Agreed

---

## Author Comment (AC2)

We sincerely thank the reviewers for their thoughtful comments, insightful questions, and constructive suggestions. Their feedback has greatly helped us clarify our ideas, strengthen our arguments, and improve the overall quality of the manuscript. Our responses are in blue text.

**Reviewer 2**

Major Comments

1. Non-invigoration aerosol-DCC interactions that could affect aerosol-ETH relationships are ignored. Aerosol-DCC interactions include direct effects on microphysics in addition to indirect effects on updraft strength. The paragraph starting on line 43 starts by referencing aerosol-DCC interactions in general but then the discussion that follows in the introduction focuses purely on updraft invigoration. This is problematic because aerosols can also directly affect microphysical properties (e.g., collision-coalescence, riming), which affects radar reflectivity and thus reflectivity echo top height. These direct effects may or may not be further associated with a change in updraft strength. To assume that updraft strength alone is the cause for changes in ETH assumes that changes in aerosols do not alter the reflectivity profile for a given cloud top. Furthermore, there is an assumption that the relationship between ETH and the true cloud top (the vertical gradient of reflectivity between the ETH and cloud top) does not change with changes in aerosols. It is not clear how valid those assumptions are. What evidence is there to suggest that ETH changes are primarily corresponding to changes in updraft strength?

We thank the reviewer for the insightful questions.

We believe that we have thoroughly described the aerosol effects on both DCC microphysics and dynamics in the introduction. This includes highlighting the high level of uncertainties in the extent of aerosol invigoration or enervation reported in the literature and the dependence of (indirect) aerosol effects on convective updrafts on the (direct) aerosol effects on DCC microphysics. The connection between direct and indirect aerosol effects is explicitly detailed through the microphysical pathways described within each invigoration mechanism outlined in the introduction.

It is indeed challenging to directly address aerosol effects on DCC microphysical properties and processes using the ARM TRACER observational datasets or any similar datasets from comparable campaigns. This is primarily because the TRACER field campaign alone did not include an in-situ cloud observational platform capable of observing quantities related to processes such as collision-coalescence and rimming.

We also acknowledge the uncertainty associated with using Echo Top Height (ETH) as a proxy for DCC intensity or maximum vertical velocity. The lack of direct measurements of convective vertical velocity remains a significant limitation, not only for this study but also for many previous observational studies. Due to such limitations, ETH has been routinely used as a proxy for DCC intensity or updraft strength in the literature (e.g., Liu and Zipser, 2013; Guo et

al., 2018; Hu et al., 2019; Veals et al., 2022). One of the advantages of using ETH here provides a means to compare our findings with prior studies and builds on the existing body of knowledge.

As the reviewer also pointed out, a detailed investigation of the correlation between cloud microphysical properties/processes and ETH is indeed a limitation of studies that rely primarily on ground-based measurements or radar retrievals in the absence of in-situ observations of DCC microphysical properties when addressing aerosol-DCC interactions. This limitation certainly applies to the manuscript under review. Even in the case of having in-situ observations of DCC microphysical and updraft properties, the causal links/structures between these quantities and ETH would be difficult to establish without the use of more advanced causal inference models or modeling components.

To address reviewer's concern, we added these paragraphs to the manuscript:

1. to Line 108 where the ETH is introduced:

"Note that this assumption neglects the possibility that aerosols may directly influence cloud microphysical processes (e.g., collision-coalescence, riming), which could, in turn, affect radar reflectivity and, consequently, the DCC ETH. Quantifying such influence is challenging in the absence of in-situ observations of the cloud microphysical and dynamical properties (e.g., hydrometeor phase/size distribution, updraft velocity). The reliance on this proxy also stems from the lack of direct measurements of convective vertical velocity for DCCs investigated here, a significant limitation not only for this study but also for many previous observational studies. Nevertheless, using ETH as a proxy allows for comparison of our findings with prior studies, which is valuable for the scientific community and for providing modeling constraints on simulations of the aerosol-DCC interactions."

2. Additionally, we have added the following text to Section 4.6 where the limitations of the study are discussed:

"In the absence of in-situ observations of cloud microphysical properties, the current analysis cannot account for any 'direct' effects of aerosols on ETH or cloud depth through microphysical processes. Neither does the study investigate the microphysical pathways through which aerosols may cause the changes in ETH. Such examinations require in-situ observations and/or high-resolution model simulations, which forms a key limitation of any study aiming to explore aerosol-DCC interactions using remote sensing retrievals alone."

2. The g-computation model does not provide the causal direction, which still needs to be assumed, even if it is called a causal inference model. This assumption is made in the multiple linear regression model where the predicted convective property is assumed to follow from the predictors. The reasoning for this is that the meteorological and aerosol properties are defined prior to the convective cell properties, which makes sense, but this is similar to what has been done in some prior studies. Furthermore, this time offset

still doesn't ensure the assumed causal direction because there is a lot of atmospheric complexity that isn't being quantified that can affect the properties of the cells and atmosphere offset in space and time. Thus, describing this research as the first to show cause-effect is misleading. The methods do have unique aspects relative to past studies that can be highlighted but there is no reason to believe that the causal direction has been more discerned than in past studies.

We acknowledge that using g-computation, like those traditional methods such as linear regression, still requires assuming a causal link between aerosol number concentration and DCC intensity. However, our approach is superior to bivariate correlations (that do not account for basic confounding effects), because it attempts to control for confounding influences of known or potential confounders. If no causal link exists between aerosols and convection, the estimated causal effect using g-computation would approach zero.

We softened our language and rewrote the last paragraph:

"Nevertheless, this study demonstrates the potential of using a causal model to evaluate the effects of aerosols on DCC properties, providing new insights into aerosol-convection interactions through observations. It also represents a step forward in addressing the challenges of disentangling aerosol-meteorological co-variability in these interactions. Additionally, this causal framework shows promise for broader applications, offering a valuable tool for exploring complex scientific questions across various disciplines."

We removed the last sentence in the abstract.

3. It is not clear what value the g-computation model provides over the multiple linear regression. If the underlying model were a more complex nonlinear model, there would be some justification for it, but multiple linear regression is used. The multiple linear regression coefficients can be used to describe convective sensitivity to aerosols, giving the same results. Even with using the g-computation model, describing an aerosol effect as just the change in ETH without the corresponding change in aerosols, as is done throughout the paper, doesn't make much sense. It is the sensitivity, i.e., the change in ETH per change in aerosol concentration, that is most relevant with the underlying assumption that this is approximately linear, and this is simply the slope for the aerosol concentration predictor from the multiple linear regression model. What does the g-computation model provide that the regression cannot other than calling the model "causal machine learning"?

We thank the reviewer for the question.

We are not denying the fact that simple linear regression can be used for causal inference, but only under ideal circumstances where individual values are *randomly assigned* to groups. This condition, however, is not applicable to our observational study or similar types of studies in atmospheric science. Fundamentally, whether simple linear regression can infer causal

relationship depends on how the data was collected. See the first few paragraphs of the introduction in Chatton et al. (2020) for more information.

In the case of aerosol-convection interactions in nature, it is *impossible* in the current world to randomly inject specific amounts of cloud condensation nuclei (CCN) into naturally developed convective clouds. The CCN concentration at a particular location on a given day can be a result of other factors, such as humidity, wind direction, and/or pre-existing convection. These hidden factors (confounders) could themselves be the true causes of changes in convective intensity. As long as CCN number concentrations cannot be randomly assigned, linear correlation coefficients cannot accurately infer causal effects, as they fail to account for basic confounding effects.

Our method extends the capability of an MLR because it offers an alternative by attempting to control for the confounding influence of known or potential confounders such as CAPE. We achieved this through a "controlled" or "forced" experiment, which, though less ideal than a fully randomized experiment, involves manipulating certain variables while others are held constant or randomized to minimize their confounding effects. In our case, we set the aerosol number concentration for every case to 1 for the polluted condition and 0 for the clean condition. Our identified confounders were kept constant in both scenarios.

Within the g-computation framework, technically, any predictive model can be used in the initial step. However, the choice of model often depends on the specific application and the number of available samples. For our study, we chose an MLR model over a different machine learning model due to the limited sample size. Note that this choice of the Q-model is NOT a direct advantage of g-computation itself; rather, the strength of this approach lies in its ability to control for confounding variables in its follow steps, which simple regression cannot achieve without the random assignment of aerosols into the atmosphere.

I am glad that the reviewer asked the question about the possibility to estimate the change in ETH per unit change in aerosol concentration. It actually is achievable using more sophisticated causal models. One such model that we have been experimenting with is called causal-curve, which allows the estimation of the causal effect of aerosols on ETH as a function of aerosol number concentration. However, this analysis is beyond the scope of the current study where we focused on estimating the *average* causal effects of aerosols. Additionally, the current manuscript is already quite lengthy, and including the description and results from the causal-curve model would make it more challenging for readers to follow. To maintain clarity and focus, we have decided to reserve this aspect for future studies.

4. Tests for multiple linear regression model accuracy and robustness are missing. For example, the predictor coefficients should have 95% confidence intervals computed. In addition, how well does the MLR predict the observed ETHs? What is its r2 value? The r2 is important as it shows how much of the ETH variance remains unexplained by the model, which is relevant for missing information that could still confound the relationships of ETH with the current predictors.

We thank the reviewer for the comments and suggestions. To address those, we included the *adjusted* R2 values for the selected exposure variables in the table below (also added in the supplemental material of the manuscript). The adjusted R2 was chosen over the R2 because it penalizes the inclusion of unnecessary independent variables. Specifically, as more predictors are added to the model, the adjusted R2 will increase only if the new variables significantly improve the model performance. In contrast, using R2, the value either remains the same or increases with the addition of new independent variables, regardless of whether the added variables significantly enhance the model performance.

As shown in the table below, the adjusted R2 values are generally below 0.5 and rarely increase even when all the potential confounders discussed in section 2.2 in the manuscript are included (according to a sensitivity test we performed). On one hand, this is, on some level, expected given the fact that these relationships are predominately nonlinear in nature. On the other hand, this result aligns with our previous statement in the manuscript: other confounding variables, beyond those included or discussed in the manuscript (section 4.6), likely exist but are unaccounted for. These variables may not have been measured or discovered to have a relationship with the outcome variables. Additionally, the small sample size may contribute to the low adjusted R2, as high variability in the outcome variable can artificially suppress it.

It is important to note that the purpose of these fitted MLR models is not to predict ETH but rather for exploration and hypothesis testing in this manuscript. Thus, the focus is on the other measures of the model robustness, making a relatively low adjusted R2 less critical. For example, in the original manuscript, we run model diagnostics (in supplemental material) to ensure the validity, reliability, and interpretability of the fitted MLR model which ensures the robustness of coefficients.

The 95% confidence intervals for the independent variables were also calculated and included in the table below. We notice that the values for the exposure variables sometimes cross 0, indicating the difficulty to conclude that the exposures have a clear and meaningful influence on the outcome. In our case, it suggests that aerosol exposure may not have a significant impact on DCC ETH. This finding is consistent with the relatively small or minimal causal effects shown for these scenarios in Figures 8 and 9.

To make this information clear, we made a few changes to the paper:

1. We modified lines 393 to 396: "We run model diagnostics to ensure the validity, reliability, and interpretability of the fitted MLR model as well as ensuring the robustness of coefficients. This is achieved by examining the key assumptions (i.e., linearity, homoscedasticity, normality, independence, and multicollinearity) of the MLR models as described in Text S4 in the supporting information. Overall, all valid scenarios presented in Section 3.2 satisfy these assumptions. In addition, we also calculated the adjusted R2 values, the 95% confidence intervals for each independent variable (Table S4 in the supporting information). The adjusted R2 values are generally

below 0.5 and rarely increase even when all the potential confounders discussed in section 2.2 are included. This result infers those other confounding variables, beyond those included or discussed here, likely exist but are not accounted for. These variables may not have been measured or discovered to have a relationship with the outcome variables which will be discussed in section 4.6. Additionally, the small sample size may contribute to the low adjusted R2, as high variability in the outcome variable can artificially suppress it."

2. We modified lines 470 to 473: "Interestingly, when conducting causal analysis on the "invalid" scenarios, the estimated average aerosol causal effects are mostly negative (Figure 8), highlighting the potential for contradictory results when a different exposure variable is used. Even for the "valid" scenarios, the significance of the estimated causal effects is challenged by the inconsistent 95% confidence intervals for the coefficients of the exposure variables in the fitted MLR models (Table S4 in the supporting information). Specifically, the 95% confidence intervals for the exposure variables sometimes cross 0, making it difficult to conclude that the exposures have a clear and meaningful influence on the outcome. This finding is consistent with the relatively small or minimal causal effects observed for these scenarios in Figures 8 and 9, which are likely to fall into the uncertainty range of the measurements or related to the sampling methods we used."

**Table**: The Adjusted $R^2$ values for the fitted MLR models and the 95% confidence intervals for the independent variables. The outcome variable is the 30 dBZ ETH and the CAPE is calculated when assuming the most unstable parcel would rise.

| Exposure, Distance to the ARM site | Adjusted $R^2$ | 95% confidence intervals for the exposure variables | 95% confidence intervals for CAPE | 95% confidence intervals for ELR3 |
|---|---|---|---|---|
| All cases | | | | |
| Ncn, 40 km | 0.2 | [-0.01  1.53] | [0.03  0.78] | [-0.02  0.76] |
| Ncn, 50 km | 0.2 | [-0.07  1.37] | [-0.02  0.69] | [0.35  1.06] |
| Nufp, 20 km | 0.1 | **[0.66 3.76]** | [-0.67  0.66] | [-1.16  0.39] |
| Nufp, 30 km | 0.2 | **[0.16  2.02]** | [-0.11  0.76] | [-0.12  0.81] |
| Nufp, 40 km | 0.2 | **[0.11  1.66]** | [0.01  0.75] | [-0.04  0.73] |
| Nufp, 50 km | 0.3 | **[0.32  1.73]** | [-0.06  0.63] | [0.32  1.01] |
| Sea breeze cases only | | | | |
| Nccn1, 30 km | 0.2 | [-0.01  4.06] | [-0.63  1.18] | [-0.65  1.32] |

| | | | | |
|---|---|---|---|---|
| Nccn1, 40 km | 0.2 | [-0.43  2.73] | [-0.44  1.02] | [-0.32  1.10] |
| Nccn1, 50 km | 0.3 | [-0.68  1.61] | [-0.00  1.06] | [0.29  1.42] |
| Nccn08, 30 km | 0.2 | [-0.01  4.06] | [-0.63  1.18] | [-0.65  1.32] |
| Nccn08, 40 km | 0.2 | [-0.57  3.01] | [-0.47  1.04] | [-0.51  1.08] |
| Nccn08, 50 km | 0.3 | [-0.70  2.05] | [-0.14  1.01] | [0.14  1.40] |
| Nccn06, 30 km | 0.2 | [-0.01  4.06] | [-0.63  1.18] | [-0.65  1.32] |
| Nccn06, 40 km | 0.2 | [-0.29  2.95] | [-0.33  1.04] | [-0.52  1.02] |
| Nccn06, 50 km | 0.4 | **[0.29  2.66]** | [-0.15  0.87] | [0.01  1.15] |
| Nccn04, 30 km | 0.3 | **[0.74  5.30]** | [-0.63  1.05] | [-1.32  0.90] |
| Nccn04, 40 km | 0.2 | [-0.57  3.01] | [-0.47  1.04] | [-0.51  1.08] |
| Nccn04, 50 km | 0.4 | **[0.06  2.62]** | [-0.23  0.87] | [0.02  1.20] |
| Nccn02, 20 km | 0.4 | **[0.75  7.44]** | [-1.07  1.20] | [-2.76  0.75] |
| Nccn02, 30 km | 0.3 | **[0.37  4.27]** | [-0.32  1.36] | [-0.79  1.17] |
| Nccn02, 50 km | 0.3 | [-0.84  1.54] | [-0.02  1.06] | [0.29  1.46] |
| Nccn01, 30 km | 0.3 | **[0.37  4.27]** | [-0.32  1.36] | [-0.79  1.17] |
| Ncn, 40 km | 0.2 | [-0.41  2.41] | [-0.17  1.15] | [-0.30  1.11] |
| Ncn, 50 km | 0.4 | **[0.22  2.26]** | [-0.12  0.89] | [0.41  1.38] |
| Nufp, 50 km | 0.4 | **[0.05  2.16]** | [-0.10  0.93] | [0.33  1.35] |

5. The argument for activation of ultrafine aerosols in updrafts leading to increases in ETHs lacks evidence. Activation of ultrafine particles seems highly unlikely given the high concentrations of larger aerosols for most of the samples assessed (Figure 7). Activation of the ultrafine particles would result in cloud droplet concentrations of a few thousand per cm3. Are there aircraft measurements (e.g., during ESCAPE) to support such high drop concentrations? Assuming a favorable composition for nucleation, what would the supersaturation need to be to activate particles at a certain size (e.g., 10 nm) given observed aerosol size distributions? This could be assessed in a parcel model to show if the argument being made is even physically possible.

We thank the reviewer for their comment. It was NOT our intent to argue that ultrafine aerosols will necessarily be activated in the updrafts. As the reviewer mentioned, there are no direct measurements of supersaturation or other indicators to definitively determine aerosol

activation within the DCCs investigated during TRACER. Our intent was to hypothesize that, IF ultrafine aerosols are activated in the updrafts, they could influence the ETH of the convection. If this activation is physically impossible under the conditions studied, it would imply that aerosols do NOT significantly affect the ETH in this particular scenario.

We softened our language and added these sentences to Line 483: "In addition, the high concentrations of larger aerosol particles observed under the assessed conditions (Figure 7) raise doubts about the likelihood of all ultrafine particles being activated. This challenges our hypothesis that aerosols may influence DCC ETH under the assumption that all ultrafine particles are activated."

6. The diurnal cycle needs to be ruled out as a cause of the CN-ETH and UFP-ETH relationships. Over land, ultrafine aerosols often have a strong diurnal cycle just as deep convection does, which can affect relationships between the two. Accumulation mode aerosols often have a much weaker diurnal cycle, which is potentially a hypothesis for why one wouldn't get robust CCN relationships but robust CN relationships with ETHs. For example, Fast et al. (2024) shows this for the CACTI campaign. This occurs because new particle formation processes over land operate during the daytime. What are the typical changes in ETH and predictor variables including CN and CCN over the diurnal cycle? Do CN and ETH variables both peak in the later afternoon? If the hour of day is controlled for, does that affect the aerosol-ETH relationships?

The reviewer raised an excellent point, and we agree that it is important to investigate whether the diurnal cycle influences the causal relationships between Ncn-ETH and Nufp-ETH. In response, we included the timing of convection initiation as an additional confounding variable in the g-computation model along with the original ones we have selected in the original manuscript.

As an example (shown in the figure below), when using 30-dBZ ETH as the outcome variable and the most unstable parcel for calculating the convective indices, the average causal effects showed very similar results. They only differ by 0.1 km compared to the scenario presented in Figure 8, where the diurnal cycle was not controlled for.

We added these sentences to Line 476 in the manuscript: "We also conducted a sensitivity test to examine whether the diurnal cycle affects the causal relationships between aerosol properties and ETH. The results indicate that the average causal effects are only 0.1 km lower than those presented in Figure 8, where the diurnal cycle was not controlled for. This suggests that the diurnal cycle has a limited influence on the aerosol causal effects on ETH under the specific environmental conditions of this study."

[Figure]

**Figure.** Average causal effects on 30-dBZ ETH estimated for each potential exposure variable after controlling for confounders which includes CAPE, ELR3, and the timing of the convection initiation. Different colors represent different maximum distances between measurements of environmental variables and DCC properties. The meteorological variables are calculated using ARM soundings (6-hr) when assuming the most-unstable parcel would rise to form a convection. The white hatch lines indicate "valid" results.

7.  Relevance of sounding convective parameters at M1 for some situations needs further inquiry. Convective parameters like CAPE are not stable for 4-6 hours over land, and the study (Prein et al., 2022) used to support this claim on line 209 does not state that so far as I can tell. That study uses a limit of 4 hours difference between observed and simulated MCSs to match them, and MCSs are not the same as isolated convective clouds in atmospheric sensitivities. Other studies such as Nelson et al. (2021) show large changes in low level moisture on distances < 50 km and times of ~1 hour over some land convective regions. The statement after this on lines 209-211 that the M1 site is not heavily affected by maritime conditions is also confusing because the M1 site is close to Galveston Bay, and as noted in the study, a bay breeze often forms. Perhaps the bay air mass is similar to the continental air mass in terms of aerosol and thermodynamic properties, but I'm not sure that can be assumed. It may not be possible to easily assess these caveats, but they should at least be highlighted. Something that could be looked into though is whether the M1 surface measurements are relevant to air feeding cells at nighttime and/or after the bay/sea breezes have passed inland of the M1 site by examining stability at and through the boundary layer up to approximate cloud base to assess the likelihood of coupling to M1 site surface conditions.

We thank the reviewer for pointing this out and agree that the lack of comprehensive spatial coverage and high-temporal-frequency measurements during similar field campaigns is one of the limitations of this type of study. While we applied a few thresholds to select suitable measurements, these are by no means perfect. Instead, they serve as an approach

to reduce uncertainty while maintaining a meaningful sample size for drawing robust conclusions.

If we were to restrict the analysis to measurements taken within one hour prior to the initiation of precipitation cores, we would be left with only nine cases (within 20 km of the site), significantly putting the robustness of the results in doubt. This highlights the need for longer-term field campaigns with improved measurement coverage to overcome these challenges, which we recommend as a priority for the broader research community.

Regarding the impact of sea breezes, we investigated their characteristics and effects in a recently published paper (Wang et al., 2024). Based on an analysis of surface variables, we observed that sea breezes have a long-lasting influence on temperature, humidity, and wind, persisting for several hours after the passage of the sea breeze front. An example showing this effect is presented in Figure 4 of Wang et al. (2024).

We rewrote lines 206 to 211: "The choice of a 6-hour time gap and a 50 km distance threshold as the upper limit represents a compromise between capturing representative environmental conditions and maintaining a sufficient sample size. We do want to emphasize the possibility of substantial temporal and spatial variability in the thermodynamic conditions around the M1 site. Local phenomena such as sea breeze, bay breeze, urban effects, and other factors may complicate the extent to which the environmental measurements at the M1 site represent the actual air mass injected into the DCCs (e.g., Rapp et al., 2024; Wang et al., 2024)."

8. More information on the spatiotemporal distribution of cells and cell properties is needed. Because of potentially substantial gradients in aerosol and thermodynamic properties given the coastal and large urban area, it would be ideal to plot the initiation locations and/or locations where the cell ETHs are maximized on maps for different ranges from the M1 site rather than the tracks in Figure 1 that don't provide much information. In addition, it would be helpful to map out cell properties like those in Figure 6 to see if there are spatial gradients in the properties with respect to the M1 site location.

We thank the reviewer for this suggestion. In response, we plotted the locations where the cell ETHs are maximized on maps for cells initiated within 50 km of the M1 site, as shown below (also included in the revised manuscript). The colors in these subplots indicate cell properties previously shown in Figure 6 of the original manuscript. Additionally, we have removed Figure 1 (cell tracks) from the original manuscript.

The spatial distribution of the cell locations reveals clustering along a line that is perpendicular to the coastline and northwest of the M1 site. One possible explanation for this pattern is the interaction between sea breeze, bay breeze, and urban heat island

induced circulations, which likely create a favorable zone for cell initiation and/or collisions (Mejia et al., 2024). These cell collisions events may lead to larger cell areas, as observed in Figure c, and slightly longer lifetimes compared to cells located farther from this zone (Figure f). This is consistent with what we found in our recent submitted paper about cell colliding and merging behaviors (Hahn et al. in review). It is not surprising that these cells tend to initiate later in the day (Figure d), coinciding with the propagation of sea and bay breezes and their convergence with urban heat island-induced circulations (Wang et al., 2024). As the reviewer also mentioned, this spatiotemporal heterogeneity introduces complexity into our study, where we rely on point measurements of environmental variables which is a practical compromise in the absence of a more comprehensive measurement network over the region. This again highlights the needs for conducting long-term ground-based field campaigns with additional instruments that cover a larger spatial range.

[Figure]

**Figure**: Dots indicate locations where the cell ETHs are maximized on maps for cells initiated within 50 km of the M1 site. The colors in these subplots indicate cell properties as shown in Figure 5.

We added these paragraphs to the revised manuscript (after Line 442): "Figure 6 illustrates the spatial distribution of DCC properties, showing a notable cluster along a line perpendicular to the coastline and northwest of the M1 site. This pattern can potentially be attributed to the interplay between sea breeze, bay breeze, and urban heat island-induced circulations, which may create a conducive environment for DCC initiation and/or collisions (Mejia et al., 2024). Such events appear to result in larger cell areas, as depicted in Figure 6c, and slightly longer lifetimes compared to cells located outside this zone (Figure 6f), consistent with findings by Hahn et al. (2024). Additionally, it is observed that these cells tend to initiate later in the day (Figure 6d), aligning with the timing of sea and bay breeze propagation and their convergence with urban heat island circulations in this region (Wang et al., 2024). Note that the spatiotemporal heterogeneity of these precipitation cores adds complexity to our study, as it relies on point measurements of environmental variables. While this approach is a practical solution given the absence of a comprehensive measurement network during TRACER, it highlights the need for long-term field campaigns with enhanced instrumentation to achieve better spatial coverage across regions with complex multiscale forcings."

9. Are ETH retrievals from level 2 NEXRAD data unbiased with range from the radar? Related to the previous comment, ETHs should be mapped with range from the radar to see if there are biases related to beam filling and gaps between elevation angles with range.

As shown in subplot (b) of the figure above, there is no clear trend indicating that ETH consistently increases or decreases with distance from the radar. Thus, we do not believe ETH is biased by radar range for cases studied here.

10. ACP recommends making processed data and code openly available in a FAIR-aligned reliable public repository to support study reproducibility. It is likely not possible to reproduce the methodology with only links to TINT and raw datasets given the information provided in the study.

We uploaded the post-processed data and codes for running g-computation to Zenodo and added a link to the revised manuscript.

1. WANG, D. (2024). Post-processed data for "Causal Analysis of Aerosol Impacts on Isolated Deep Convection: Findings from TRACER in Houston-Galveston". Zenodo. https://doi.org/10.5281/zenodo.14298966

2. WANG, D. (2024). Codes for "Causal Analysis of Aerosol Impacts on Isolated Deep Convection: Findings from TRACER in Houston-Galveston". Zenodo. https://doi.org/10.5281/zenodo.14299094

**Minor Comments**

1. Line 7: Only a single model predicts a significant relationship between an aerosol concentration and convective core area, which is 0.8% CCN within 30 km of the M1 site (Figure 10). The other 31 models are not significant. That seems pretty random, particularly since some models switch sign with changes in range within M1, and not enough to support this statement in the abstract that greater aerosol levels correspond to increased convective core area.

We removed the area statement in the abstract.

2. Lines 31-33: This is an odd motivation since ERFaci uncertainty is currently mostly attributed to non-deep convective clouds that are not the focus of this study.

We change the sentences to: "Aerosol-cloud interactions in DCCs are among the most complex and challenging processes to simulate accurately. This difficulty was evidenced in a recent model intercomparison project (MIP) conducted by the Deep Convective Working Groups of the Aerosols, Cloud, Precipitation and Convection (ACPC) initiative (Marinescu et al., 2021)."

3. Discussion of leading invigoration mechanisms in introduction: Semi-direct effects by aerosols that alter atmospheric thermodynamic stability should also be included.

We thank the reviewer for the suggestion and agree that a discussion of the semi-direct effect could provide additional context. However, considering that the introduction and manuscript are already quite long, and our work focuses specifically on the evidence of invigoration and enervation, we prefer to limit the discussion to these effects. Moreover, investigating the semi-direct effect would be equally challenging for us due to the lack of information on the extent of aerosol mixing into clouds and its contribution to this effect, as well as the absence of direct cloud microphysical measurements required to study it comprehensively.

4. Lines 60-63: Some of the studies cited here are not simply questioning the importance of invigoration mechanisms relative to other forcings but showing that there is a spectrum of enervation to invigoration possible, thus suggesting that referring to the mechanisms only in terms of invigoration is misleading.

We rewrite these sentences: "Despite a range of hypothetical mechanisms for aerosol-DCC invigoration, recent studies continue to challenge these theories, revealing a spectrum ranging from enervation to invigoration (e.g., Grabowski and Morrison, 2020; Igel and van den Heever, 2021; Dagan, 2022; Romps et al., 2023; Peters et al., 2023)."

5.  Lines 75-76: Though individual modeling studies have quantified aerosol effects, it is important to note that there is still disagreement between these studies, even in the sign of effects, because models and the methods for analyzing them (e.g., discussion in Varble et al., 2023).

We added these sentences to Line 76: "Though individual modeling studies have quantified aerosol effects on DCCs, it is important to note that there remains significant disagreement between these studies, even in the sign of effects, largely due to variations in model configurations and the methods used to analyze them (Varble et al., 2023)."

6.  It isn't clear how updraft strength is being defined. Is this referring to updraft mass flux, average vertical wind speed, or maximum vertical wind speed?

In this context, we are referring to the maximum vertical velocity in convective regions. We changed this sentence to "The maximum height of these cores can serve as a proxy for the maximum updraft velocity…"

7.  Lines 124-128: Not tracking cells when max 2-km Z < 40 dBZ leaves out more than non-precipitating stages as suggested here. It also leaves out lightly precipitating periods.

We rewrote the sentence: "In other words, the tracked lifetime of the cores excludes the initiation stage of non-precipitating cumulus clouds, the dissipation stage of non-precipitating anvil clouds, and the lightly precipitating periods during either stage."

8.  For the meteorological variables, there is almost an unlimited number that could potentially be relevant and tested. Were different shear layers other than 0-5 km tested? Was mid-level RH tested (separate from the boundary layer)?

We did a sensitivity test by adding mid-level RH (3-6 km) and high-level wind shear (5-10 km) as confounding variables alongside CAPE and ELR3 to the g-computation model. We found a decreased aerosol average causal impact on 30 dBZ ETH by less than 0.1 km, which is minimal.

9.  What assumptions are made for the lifted parcel calculations (LCL, LNB, CAPE)? Is liquid pseudoadiabatic or reversible ascent assumed?

We added one sentence to line 160: "Note that, in the calculations, we assume that the parcel undergoes undiluted ascent in a pseudo-adiabatic process (neglecting hydrometeor loading)."

10. Line 187: CCN at various supersaturations does not have a temporal resolution of 1 minute or less as stated here. The supersaturation is varied over the course of about an hour usually so there is 1 value at each supersaturation every ~hour or so.

We rewrote this sentence: "The Ncn and Nufp were measured at a temporal resolution of 1 minute, Nccn at various SSs had two measurements per hour, and radiosondes, used to derive meteorological parameters, were launched four to seven times per day."

11. Lines 194-195: A t-test may not be valid here if the aerosol distributions are skewed.

A t-test is valid if the sample size (n) is large enough. The general rule of thumb is n>30. According to the Central Limit Theorem, the *mean* of any distribution is approximately normally distributed when the sample size is sufficiently large. In other words, even if the data itself is not normally distributed, the mean of the data *is*. If we were to repeat the experiment (TRACER field campaign) a hundred times and plot the sample *means*, the resulting distribution would be approximately normal.

12. How are DCC tracking results averaged? Does each DCC have a single value for a variable like ETH and then all of the ETHs are averaged together?

Yes, we took the maximum ETH throughout a tracked DCC lifetime (so one ETH for one DCC), then we averaged these ETHs to represent the mean ETH of these qualified DCCs for each corresponding sounding.

We added these sentences to line 204: "More specifically, in terms of ETH, we identify the maximum ETH throughout a tracked DCC lifetime (one ETH for one DCC), then we average these ETHs to represent the mean ETH of these qualified DCCs."

13. Lines 234-235: I don't follow the argument for why large-scale ascent needs to be avoided, though I can see why MCSs would want to be avoided. Is that the primary reason for avoiding certain large-scale meteorological conditions?

Yes, mostly. We want to eliminate large-scale, dynamically-driven convective clouds since the aerosol effect may be overwhelmed by meteorological forcing. These situations often also exhibit strong CAPE, in which the aerosol effects are found to be difficult to detect (Storer et al., 2010). Additionally, different types of convection (organized vs. isolated) may respond to aerosol loading differently (Chakraborty et al., 2016). Therefore, we want to focus our study only on isolated convective clouds that are initiated in a similar weak large-scale forcing environment, where the aerosol effect may be more identifiable.

We added this sentence to line 235: "This choice serves to mitigate the potential influence of large-scale ascent on the evolution of DCCs. In other words, we aim to exclude large-scale, dynamically-driven convective clouds, such as mesoscale convective systems, since the aerosol

effect may be overwhelmed by meteorological forcing (Chakraborty et al., 2016; Storer et al., 2010)."

14. Lines 273-275: Mesoscale deep convective systems are still buoyancy driven, so I don't understand what this sentence is trying to get across.

We removed this sentence and rewrote the previous one (see answers to question 13) to emphasize that we are only considering isolated convective clouds, which may be more conducive to observing aerosol impacts.

15. Figure 4: Why are values not filled in for the significant correlations less than 0.4? Also, I may have missed it, but are the aerosols in Figure 4 sampled around the same time as the soundings or are they sampled after the soundings?

Given the size of the correlation matrix, including correlation coefficients below 0.3 makes it difficult to visually extract the most important information. Since relationships with smaller correlation coefficients are not the focus of this study, we have only included values greater than 0.4 to improve the readability of the figure.

The post-sounding averaging (a 1-hour period following the radiosonde launch) for aerosols is shown in Figure 4. We added this information to the figure caption. We also plotted the correlations using the prior-rain averaging method (a 1-hour period before the rain). The resulting correlation coefficient matrix is very similar to the one shown in Figure 4, and we have included it in the supplemental materials.

16. In some places, LWS is used and in others, shear is used. It would be best to choose one or the other and be consistent throughout.

We modified the text and used LWS throughout the manuscript.

17. Line 342: Should "accuracy" be "robustness" here?

We changed it to "robustness" in the revised manuscript.

18. Lines 364-365: Including some critical meteorological quantities supports this assumption, but I wouldn't say that it is necessarily sufficient. That is hard to know without an in-depth study of possible confounders.

The reviewer is correct. However, as much as we would like to include all possible confounders, we often need to consider the balance between the number of samples and the number of confounders. An exhaustive list of confounders is ideal, but it may come at the cost

of model accuracy and stability when we work with a limited sample size. This is a challenge the community faces today when working with observational data from shorter-duration campaigns, especially when certain data streams are only available at a single location.

We changed the sentence to: "Critical quantities known to influence ETH, such as CAPE and LWS, are explicitly included or discussed, to a large extent, supporting this assumption."

19. Lines 388-389: I don't follow the argument of multi-collinearity supporting standardization. Isn't the reason for standardization stated on lines 390-392?

Yes, it does not directly prevent multicollinearity but can prevent the numerical instability in computations that might arise when multicollinearity exists. We removed that sentence from the manuscript to avoid confusion. We did a test for multicollinearity by assessing the Variance Inflation Factor (VIF), and for all the models, the values are around 1, showing no multicollinearity. This information was included in the original supplemental material.

20. Lines 463-464: There is not enough evidence to make this statement that Ncn and Nupf are causing higher ETH via their activation.

We would like to direct the reviewer to our responses to major comments #5.

21. Line 494: I disagree that a causal link was demonstrated. The only supporting cause is that the aerosols are sampled prior to cells in time, but there is no evidence to show the causal mechanisms, and there are potentially other confounders not accounted for (see major comments).

We would like to direct the reviewer to our responses to major comments #2 and #3.

22. Lines 536-540: It's true that uncertainty renders the max reflectivity results less robust, but the same argument can be made for how well 4-6 hourly soundings and aerosols at a single point represent conditions where cells are growing.

We emphasized this uncertainty source in the second paragraph located in Section 4.2 in the revised version.

23. Lines 603-605: I think this sentence can be clarified. Aerosol is not *robustly* associated with DCC max ETH (not its evolution) given the sampling in this study. That does not mean that it couldn't be if more samples were added.

We rewrote the sentence: "Only a small fraction (16%) of the SLR models are valid, indicating that, in the majority of cases, aerosol loading is not robustly associated with DCC maximum

ETH, suggesting insufficient effects of aerosols on DCC updraft velocity in these situations with the current sample sizes."

**References**

Fast, J. D., Varble, A. C., Mei, F., Pekour, M., Tomlinson, J., Zelenyuk, A., Sedlacek III, A. J., Zawadowicz, M., and Emmons, L. K.:, 2024 Large Spatiotemporal Variability in Aerosol Properties over Central Argentina during the CACTI Field Campaign, EGUsphere [preprint], https://doi.org/10.5194/egusphere-2024-1349.

Nelson, T. C., J. Marquis, A. Varble, and K. Friedrich, 2021: Radiosonde Observations of Environments Supporting Deep Moist Convection Initiation during RELAMPAGO-CACTI. *Mon. Wea. Rev.*, 149, 289–309, https://doi.org/10.1175/MWR-D-20-0148.1

Citation: https://doi.org/10.5194/egusphere-2024-2436-RC2

We cited these papers in Section 4.3 in the revised manuscript.

---

## Author Response (AR2)

**Responses to Reviewers:**

We thank the reviewer for their insightful comments and apologize for any confusion caused by our initial description in the manuscript and responses. We appreciate the opportunity to clarify and enhance our explanation and manuscript.

Major Comment

I do not understand the reply to my major comment 3 regarding whether g-computation and multiple linear regression results differ. From my read of section 3.4 describing the g-computation model and the Zenodo-archived code, the g-computation model produces the same output as the underlying multiple linear regression (the "Q-model"). If the multiple linear regression model is being used to estimate the aerosol effect on ETH, that is perfectly fine. However, if that is the case, the language in the study needs to be changed throughout it because it is stating that a new causal model is being used that is superior to past studies using regressions or other predictive models in isolating causal effects. From what I can tell, this isn't true, but it is possible I am missing something. If I am, then I suspect others will too because it is not clear from the paper or the response to my comment. Thus, I think it is important that the authors clearly demonstrate how the result from the g-computation model differs from that obtained from the multiple linear regression model alone with A=1 and A=0 inputs, which just produces the b1 coefficient being multiplied by A, as shown below.

As stated in section 3.4, a multiple linear regression (the Q-model) is performed with standardized variables (Y = b0 + b1*A + b2*V1 + b3*V2). Y is ETH, V1 and V2 are meteorological confounders, and A is aerosol concentration, where observations are inputted to derive the b coefficients. Then, 0 is used as an input for A to represent clean conditions and 1 is used as an input to represent clean conditions with V1 and V2 either held constant or entered as observed values for each observations (this isn't entirely clear, but shouldn't matter so long as V1 and V2 inputs are the same for both the clean (A=0) and polluted (A=1) calculations). The 2 multiple linear regression calculations are differenced to give a change in Y (ETH). Since A is standardized, A = 0 is the same as the mean aerosol concentration and A = 1 is 1 standard deviation above the mean and the change in Y (ETH) is associated with a 1 standard deviation change in A (aerosol concentration). Is this what is being done? If so, then this would just give b1 as the answer, which is the sensitivity of Y (ETH) to A (aerosol concentration). Mathematically, holding V1 and V2 to constant values or using the values from any given event to control for confounding variables, this would be:

Y(A=1) − Y(A=0) = (b0 + b1 + b2*V1 + b3*V2) − (b0 + b2*V1 + b3*V2) = b1

Weighting this result by polluted vs. clean conditions still gives b1 since it is a constant, and the ETH change becomes the change per 1 standard deviation change in aerosols (A=1 minus A=0) holding confounding factors the same for each scenario. Note that b1 is easier to interpret than the ETH change that is provided now because it is an ETH change per

aerosol concentration change. This is purely obtained from the multiple linear regression model and similar to past studies using multiple linear regression to estimate causal effects. Perhaps something else is being done, but this is how the paper and code currently read.

To clarify how calculations are being done, I suggest writing out the math so that it is clear how the ETH change is obtained to conclusively show one way or the other whether anything other than the multiple linear regression is being used. If only the regression is being used, then the paper should remove reference to g-computation and a new causal inference framework that is superior to past regression or random forest approaches. If instead something is being done that produces a different result than the regression alone, then that needs to be clearly demonstrated.

The reviewer is correct in recognizing that the coefficient *b1* from the standard multiple linear regression (MLR) model coincides with the Average Causal Effect (ACE) estimated through g-computation in the original manuscript. Like many models, g-computation is not without limitations, and using a simple, standard MLR model as the Q-model can obscure the inherent advantages of this causal framework, making the result quantitatively indistinguishable from *b1*, despite their *fundamental differences*.

In our response below, we demonstrate how employing more complex Q-models removes this coincidence. We therefore encourage readers to use more sophisticated Q-models when applying g-computation (if sample size allows).

In addition, the flexible nature of g-computation allows for the incorporation of machine learning models for resolving *non-linear* relationships within the data - something not achievable with standard MLR.

The reviewer's feedback also motivated us to explore a model-ensemble approach, where we report ACE distributions from g-computation using multiple Q-models for each scenario. This, too, is not feasible with MLR alone. We believe that this approach enhances the robustness of our findings, reduces biases associated with relying on a single model, and provides uncertainty estimates (which is rarely done in observational studies on aerosol-convection interactions).

We explain in detail below:

**1. Results When Using Different Q-models**

In the revised manuscript, we eliminated the use of the standard MLR and introduced three other Q-models to demonstrate the advantages of the g-computation and reveal the difference between *b1* and ACE. It also helps prevent confusion between standard MLR and g-computation.

**1.1. First Q-Model: MLR with Interaction Terms**

To *intuitively* illustrate how the result from g-computation differs *quantitatively* from the MLR coefficient *b1* (equivalent to $\beta_1$ in the equations below), we still use a MLR model but

with interaction terms to account for potential interactions between the exposure variable and confounders.

The outcome *Y* can be expressed as:

$$Y = \beta_0 + \beta_1 X + \beta_2 A + \beta_3 B + \beta_4 (A \cdot X) + \beta_5 (B \cdot X) \quad (1)$$

Here, *X* is the exposure variable, *A* and *B* are confounders, $\beta_0$ is the intercept, $\beta_1$, $\beta_2$, $\beta_3$, $\beta_4$, $\beta_5$ are coefficients, and *A·X* and *B·X* are the interaction terms.

Next, we generate counterfactual outcomes by setting the exposure variable *X* to fixed values representing different scenarios. As we demonstrate below, this step is critical for estimating the ACE and clearly sets this methodology apart from using MLR alone. We first forcefully set *X* = 1 for each individual case in the data for the polluted condition, and the potential outcome value that would have been observed is expressed as:

$$Y|do(X = 1) = \beta_0 + \beta_1 + \beta_2 A + \beta_3 B + \beta_4 A + \beta_5 B \quad (2)$$

We then forcefully set *X* = 0 for each individual case in the data to calculate the potential outcome that would have been observed if every case occurs in a clean condition:

$$Y|do(X = 0) = \beta_0 + \beta_2 A + \beta_3 B \quad (3)$$

Therefore, the expected ACE of *X* on *Y* can be expressed as:

$$ACE = E[Y|do(X = 1) - Y|do(X = 0)] = \beta_1 + \beta_4 E[A] + \beta_5 E[B] \quad (4)$$

, where E[*A*] and E[*B*] are the expected values of *A* and *B*, represented as the population means of confounders *A* and *B*.

Due to the inclusion of interaction terms, the ACE (calculated using Equation 4 and shown in the figure below) *no longer* equals $\beta_1$ (*b1* originally), unlike in the simpler MLR example from our original manuscript. Although the differences between this ACE and the one presented in our original manuscript are within ±0.1 km (which may seem negligible for this study), g-computation uncovers causal effects rather than merely representing associations between variables, as MLR does.

We replaced the standard MLR with the Equation 1 in the revised manuscript and explicitly listed all equations above to help readers better understand g-computation.

[Figure]

**1.2. Second Q-Model: Elastic Net Regression**

The second Q-model we tested is the Elastic Net regression (Zhou and Hastie, 2005), an extension of linear regression that incorporates both Lasso and Ridge regularization penalties (two widely used regularization techniques in machine learning) into the loss function. This approach helps prevent overfitting and improves generalizability, which is especially important for small datasets like ours, where MLR models may struggle with overfitting. Note that the model equation for **Y** remains a MLR equation as shown in Equation 1 (with interaction terms), but the regularization penalties affect how the coefficients ($\beta_0$, $\beta_1$, $\beta_2$,...) are estimated.

The loss function can be expressed as:

$$L = \frac{1}{2n}\sum_{i=1}^{n}(y_i - \hat{y}_i)^2 + \lambda[\alpha \sum_{j=1}^{p}|\beta_j| + (1-\alpha)\sum_{j=1}^{p}\beta_j^2] \quad (5)$$

The first part, $\frac{1}{2n}\sum_{i=1}^{n}(y_i - \hat{y}_i)^2$, represents the mean squared Error (MSE), measuring the prediction error which is the difference between predicted ($\hat{y}_i$) and actual values ($y_i$) (same as in MLR). The second part is the penalty term, which includes both the Lasso penalty ($L_1$) and the Ridge penalty ($L_2$). The Lasso penalty, $\sum_{j=1}^{p}|\beta_j|$, adds the sum of absolute values of coefficients as a penalty term and encourages sparsity by pushing some coefficients to zero. It helps reduce model complexity by eliminating irrelevant features/predictors. Meanwhile, the Ridge penalty, $\sum_{j=1}^{p}\beta_j^2$, adds the sum of squared coefficients as a penalty term and shrinks all coefficients, reducing their magnitude without setting them to zero. It encourages small, nonzero coefficients, reducing the impact of multicollinearity. Note that $n$ represents the total number of training examples (data point) in the dataset, while $p$ represents the total number of predictor variables (features) in the model.

The parameter α is the mixing parameter (ranging from 0 to 1) that determines the balance between $L_1$ and $L_2$ regularization. For α = 0, the penalty is purely from Ridge regularization ($L_2$) and for α = 1, the penalty is purely from Lasso regularization ($L_1$). For 0 < α < 1, the

penalty is a combination of both $L_1$ and $L_2$, called Elastic Net regularization. There is no definitive way to choose this parameter, in our analysis, we set α to 0.5.

The parameter λ controls the strength of the regularization/penalty. A higher λ value increases the penalty, forcing coefficients $\beta_j$ to be smaller, reducing overfitting. We fine-tune λ using a 5-fold cross-validation for each considered scenario separately (each bar in the figure below). Our results show that λ ranges from 0.01 to 10, depending on the scenario.

The ACE estimated using g-computation with Elastic Net regression as the Q-model is shown in the figure below. While the sign of the ACE for each scenario remains mostly the same as in the figure above, the overall ACE values are generally smaller, with some dropping to zero in certain scenarios. These zero ACE scenarios indicate that aerosols have no effect on ETH under these conditions. Note that the ACEs presented in the figure below also no longer equal $\beta_1$ (b1 in the original manuscript).

Section 4.3 of the revised manuscript presents the results from the Elastic Net regression Q-model, with details on the model setup and parameters provided in the supplemental material.

[Figure]

**1.3. Third Q-Model: Support Vector Regression (SVR)**

We further explore a non-linear Q-model using SVR (Smola & Schölkopf, 2004). SVR is a supervised machine learning technique designed to find a function such that most data points lie within an $\epsilon$-tube, meaning their predicted values deviate at most $\epsilon$ from the true values. In other words, SVR aims to fit the data within a specified margin of tolerance ($\epsilon$), balancing smoothness with accuracy. This approach penalizes only large errors that exceed $\epsilon$, while small deviations are allowed.

For a training set $T = \{(X_i, y_i)\}_{i=1}^l$, where $X_i \in R^N, y_i \in R$, the SVR function can be expressed as:

$$f(X) = w^T \cdot \phi(X) + b \quad (6)$$

where $\mathbf{w} \in R^N$ is the coefficient vector in feature space, $b \in R$ is the intercept, and $\phi(.)$ denotes the kernel function that maps the input $\mathbf{X}$ to a vector in the feature space. SVR supports multiple kernel functions (e.g., linear, polynomial, or Radial Basis Function) to model diverse data patterns. In our study, we use the Radial Basis Function kernel to capture potential non-linear relationships in our data.

The solution of $\mathbf{w}$ and $b$ can be obtained by solving the optimization problems:

$$\text{Minimize} \ \frac{1}{2}||w||^2 + C \sum_{i=1}^{n}(\xi_i^+ + \xi_i^-) \quad (7)$$

This is subject to the following constraints:

$$y_i - f(X_i) \le \epsilon + \xi_i^+ \quad (8)$$

$$f(X_i) - y_i \le \epsilon + \xi_i^- \quad (9)$$

$$\xi_i^+, \xi_i^- \ge 0 \quad (10)$$

Here, the first term, $\frac{1}{2}||w||^2$, controls model complexity (smaller weights lead to a simpler model). The second term, $C \sum_{i=1}^{n}(\xi_i^+ + \xi_i^-)$, measures the sum of slack variables ($\xi_i^+, \xi_i^-$), which account for deviations beyond the margin $\epsilon$. This term is introduced because a perfect fitted function $f(X)$ in Equation 6 with $\epsilon$ precision may not exist or feasible (Cortes and Vapnik, 1995). In other words, these slack variables allow the model to tolerate some degree of error in the fitting process.

The parameter C (> 0) in Equation 7 regulates the trade-off between $f(X)$ complexity and the degree to which deviations from $\epsilon$ are tolerated. If C is too high, the model penalizes errors heavily and may overfit, trying to fit all points exactly; if too low, the model ignores smaller deviations and focuses on capturing general trends, leading to a simpler model. In our study, we set C = 1.0, which represents moderate regularization, balancing error minimization with model complexity.

The parameter $\epsilon$ defines a margin of tolerance within which errors are ignored; the model does not penalize errors smaller than $\epsilon$. If $\epsilon$ is too small, the model captures more details but risks overfitting; if too large, it may miss significant variations in the data. We use 5-fold cross-validation to evaluate different $\epsilon$ values for each scenario and select the one that minimizes the mean squared error. Depending on the scenario, $\epsilon$ ranges from 0.1 to 1 in our study.

The results from the SVR-based Q-model are presented in the figure below. Overall, the estimated ACEs are smaller compared to those in our original manuscript. For a portion of

the scenarios, the values are reduced by half compared to the original results. However, the overall sign of the ACE remains consistent with the results from other Q-models.

Section 4.3 of the revised manuscript presents the results from the SVR-based Q-model, with details on the model setup and parameters provided in the supplemental material.

[Figure]

We present box-whisker plots to show the distributions of ACEs for all scenarios based on the three Q-models when using different exposure variables, as shown in the figure below. The differences are evident across the Q-models when both 30 dBZ and 15 dBZ are used as outcomes. Using MLR models (with interaction terms) overestimates the impacts (both positive and negative) of aerosols compared to the other two models. This result further reveals the limitation of aerosol impacts on DCC intensity.

[Figure]

*Caption: Average causal effects on ETH estimated using three Q-models. The post-sounding aerosol averaging period is considered. Scenarios with different radii from the ARM sites are all included. The meteorological variables are calculated using ARM soundings (6-hr) when assuming the most-unstable parcel would rise to form a convection.*

It is important to note that our candidate Q-models are somewhat constrained by the small sample size in this study. With a larger dataset, more complex Q-models, including a variety of machine learning-based predictive models, could be employed. The same subsequent steps can then be followed to calculate ACE. An ensemble modeling approach may be particularly useful in such cases (as shown in the figure above) to provide uncertainty estimates and enhance the robustness of the results. This is also critical for research on aerosol-deep convection interactions, where relying on a single model may introduce biases that skew conclusions in one direction or another (invigoration, enervation, or no effect). Employing multiple Q-models provides a way to quantify uncertainties and encourages researchers to use non-linear models and move beyond simple MLR.

Although we have demonstrated that the results from g-computation can be quantitatively different from those of MLR, we want to strongly emphasize that the two methods are *fundamentally different* in their ability to resolve true causal effects. MLR does NOT inherently account for confounding or causal structure, which is a key capability of g-computation. This distinction is particularly important in aerosol-deep convection interaction studies, where meteorological factors can often buffer aerosol effects (Stevens and Feingold, 2009). Properly accounting for confounders is essential for obtaining more accurate estimations of aerosol impacts on deep convection.

**2. Summary of the differences between G-Computation and MLR**

We recognize that when using a standard MLR, as in our original manuscript, the regression coefficient of the exposure variable quantitatively aligns with the results from g-computation. However, this does not mean that MLR and g-computation are equivalent; in fact, their purposes and interpretations are fundamentally different. Moreover, it also does not imply that MLR can be used for estimating causal effects without specific constraints or conditions. Additionally, g-computation is just one of many causal inference methods. If other causal inference methods/frameworks (e.g., propensity score matching [Rosenbaum et al., 1983]) were employed (not relying on fitting a Q-model), the results would probably differ between the two approaches (Chatton et al., 2020).

Basically, MLR is a statistical tool for modeling the relationship between a dependent variable and multiple predictors. Its primary purpose is to predict outcomes and estimate *associations*, not causal effects. It does *not* inherently account for confounding, and its regression coefficients only represent causal effects under *strict assumptions* (e.g., no unmeasured confounding, correct model specification, random exposure assignment).

G-computation, on the other hand, is a powerful causal inference method that explicitly estimates the causal effect of an intervention by modeling the relationship between variables and simulating *counterfactual* outcomes. G-computation controls for confounding by keeping confounding variables constant across hypothetical interventions (i.e., setting all exposure to 1 or 0). This ensures that the estimated causal effects are not biased by confounders, leading to a more accurate assessment of the true causal effect of the exposure on the outcome. Its flexible framework allows for the incorporation of advanced *non-linear* predictive models as Q-models to improve reliable and robustness of the estimation.

Finally, we encourage the use of more complex Q-models and a model-ensemble approach to fully leverage the advantages of g-computation.

**3. Changes to the revised manuscript**

We replaced the simple, standard MLR model with the first Q-model we presented in this document, and we added the Equations (1) - (4) to the revised manuscript (in section 3.4) to better explain the calculation. We uploaded the code to Zenodo. We also updated figures 8 - 11 and Tables 3 and S4 using results from new Q-models and model ensembles. We added the box-whisker plot shown above to the revised main manuscript as Figure 12. We also explicitly explain each additional Q-model and their parameter setting in the supplemental material. We made minor edits throughout the manuscript to reflect these changes.

Particularly, we added these paragraphs to the section 4.3 in the revised manuscript and add details of each Q-model to the revised supplemental materials:

*"The flexible nature of g-computation allows for the incorporation of advanced predictive models, such as machine learning models, to capture non-linear relationships within the data. Therefore, we employ the Support Vector Regression (SVR) model (Smola & Schölkopf, 2004) as the Q-model to examine whether the results change. Additionally, we explore a model-ensemble approach by reporting statistical results from g-computation using multiple Q-models for each scenario. To achieve this, we add results from the Elastic Net Regression Q-model to provide uncertainty estimates. We believe this approach enhances the robustness of our findings and reduces uncertainty associated with relying on a single model. This is also critical for research on aerosol-DCC interactions, where relying on a single model may introduce biases that skew conclusions in one direction or another (invigoration, enervation, or no effect).*

*SVR is a supervised machine learning technique designed to find a function such that most data points lie within an $\epsilon$-tube, meaning their predicted values deviate at most $\epsilon$ from the true values (Vapnik, 2013). In other words, SVR aims to fit the data within a specified margin of tolerance ($\epsilon$), balancing smoothness with accuracy. This approach penalizes only large*

*errors that exceed $\epsilon$, while small deviations are allowed. It is suitable for our small sample size. The detailed model and parameter settings are described in the supplemental material.*

*Elastic Net regression (Zhou and Hastie, 2005) is an extension of linear regression that incorporates both Lasso and Ridge regularization penalties (two widely used regularization techniques in machine learning) into the loss function. This approach helps prevent overfitting and improves generalizability, which is especially important for small datasets like ours, where MLR models may struggle with overfitting. Note that the model equation for Y remains a MLR equation as shown in Equation 1 (with interaction terms), but the regularization penalties affect how the coefficients ($\beta_0$, $\beta_1$, $\beta_2$,...) are estimated. We present the detailed model and parameter settings in the supplemental material.*

*We show box-whisker plots to illustrate the distributions of aerosol causal effects based on all three Q-models when using different exposure variables in Figure 12. The differences are evident across the Q-models when both 30 dBZ and 15 dBZ are used as outcomes. Using MLR models (with interaction terms) overestimates the impacts (both positive and negative) of aerosols compared to the other two models. This finding underscores the uncertainties in aerosol effects on DCC intensity and highlights the advantages of an ensemble modeling approach for providing uncertainty estimates and enhancing result robustness. This is particularly critical for aerosol–deep convection interaction research, where reliance on a single model may introduce biases that skew conclusions toward invigoration, enervation, or no effect.*

*Note that when using a standard MLR model (without interaction terms) as the Q-model for g-computation, the regression coefficient of the exposure variable from the standard MLR aligns quantitatively with g-computation results, potentially obscuring the latter's inherent advantages. However, this coincidence does not imply equivalence between MLR and g-computation; their purposes and interpretations are fundamentally different. Moreover, it does not suggest that MLR can estimate causal effects without specific constraints (e.g., no unmeasured confounding, correct model specification, random exposure assignment). To fully leverage g-computation's advantages, we encourage the use of more complex Q-models and a model-ensemble approach."*

Minor Comment

Related to my previous major comment 1, I still feel that the introduction focuses primarily on aerosol effects on cloud dynamics. This is fine if the authors prefer to keep it this way, but why I had recommended broadening the discussion to direct aerosol effects on cloud microphysics is that throughout the introduction, there are references to aerosol-DCC interactions in general. For example, lines 31 and 41 begin paragraphs by using this phrase but then just focus on indirect effects on dynamics. Aerosol-DCC interactions is not synonymous with aerosol indirect effects on cloud dynamics, so I think it needs to be clearer

when mentioning aerosol-DCC interactions that the manuscript specifically focuses on aerosol effects on dynamics rather than direct effects of aerosols on hydrometeor properties or convective cloud effects on aerosols.

We agreed and we modified the first sentence of lines 31 and 41 to highlight that we are particularly interested in convective dynamics other than other aspects of the aerosol–cloud interactions in DCCs.

*"Aerosol–cloud interactions in DCCs, particularly the aerosol effects on convective dynamics, are among the most complex and challenging processes to simulate accurately. Note that this study and its introductory discussion mainly focus on aerosol effects on convective dynamics."*

---

## Author Response (AR3)

**Responses to Reviewers:**

We would like to thank the reviewer for their thoughtful comments.

We have revised the manuscript based on the suggestions, including the removal of the g-computation approach and the two additional Q-models.

---

## Author Response (AR4)

**Responses to the Reviewer**

Removal of g-computation has made the manuscript much easier to follow without changing conclusions. The paper is acceptable for publication following clarification of one remaining issue. In previous manuscript versions, MLR predictors were standardized ((predictor – mean)/standard deviation). For a standardized aerosol predictor (X), X = 0 was the mean and X = 1 was 1 standard deviation higher than the mean even though 0 was described as clean (rather than the mean) and 1 as polluted. This most recent manuscript version simply states that X is transformed into a binary distribution of 0 or 1 without saying how this is done. The figures have not changed, so does 0 and 1 still refer to inputs to standardized predictors? If so, then 0 does not represent clean conditions but mean conditions so the difference between X=1 and X=0 scenarios represents the effect of a 1 standard deviation change in X rather than the difference between clean and polluted. If X is not a standardized variable, then how are 0 and 1 determined?

We apologize for the miscommunication. We accidentally removed the texts about standardization in the last version.

We performed standardization only on the confounding variables (e.g., CAPE, ELR), *not* on the exposure variable (aerosol predictors). The aerosol predictors were simply transformed into a binary distribution of 0 or 1 in both the with-g-computation and without-g-computation versions, *without standardization*. In other words, the transformation in both article versions is consistent. We assign the aerosol number concentration that is higher than the median value to 1 and lower to 0, as written in lines 323-325 in the manuscript. Therefore, our results remain unchanged.

We added these sentences to the revised manuscript: "In addition, we perform standardization on the confounding variables. This standardization process transforms the variables so that they have a mean of 0 and a standard deviation of 1. It is achieved by subtracting the mean of each variable from each observation and then dividing by its standard deviation."